# Beyond Task-Specific Reasoning: A Unified Conditional Generative Framework for Abstract Visual Reasoning

Fan Shi [1]   Bin Li [* 1]   Xiangyang Xue [1]

## Abstract

Abstract visual reasoning (AVR) enables humans to quickly discover and generalize abstract rules to new scenarios. Designing intelligent systems with human-like AVR abilities has been a long-standing topic in the artificial intelligence community. Deep AVR solvers have recently achieved remarkable success in various AVR tasks. However, they usually use task-specific designs or parameters in different tasks. In such a paradigm, solving new tasks often means retraining the model, and sometimes retuning the model architectures, which increases the cost of solving AVR problems. In contrast to task-specific approaches, this paper proposes a novel Unified Conditional Generative Solver (UCGS), aiming to address multiple AVR tasks in a unified framework. First, we prove that some well-known AVR tasks can be reformulated as the problem of estimating the predictability of target images in problem panels. Then, we illustrate that, under the proposed framework, training one conditional generative model can solve various AVR tasks. The experiments show that with a single round of multi-task training, UCGS demonstrates abstract reasoning ability across various AVR tasks. Especially, UCGS exhibits the ability of zero-shot reasoning, enabling it to perform abstract reasoning on problems from unseen AVR tasks in the testing phase.

## 1. Introduction

Abstract visual reasoning (AVR) is a fundamental cognitive ability that enables humans to identify visual concepts, infer underlying abstract rules, and generalize the rules to new scenarios effectively (Cattell, 1963; Zhuo & Kankanhalli,

2021; Małkiński & Mańdziuk, 2022a). This ability is pivotal in cognitive activities like decision-making, visual analogy and spatial reasoning (Gray & Thompson, 2004; McCloskey, 1983). Many psychological tests are designed to assess the AVR ability of humans, which often requires the subject to discover specific rules that connect different visual concepts (Raven & Court, 1938; McMillen, 2007; Nie et al., 2020). For example, Raven's Progressive Matrix (RPM) (Raven & Court, 1938) focuses on assessing generalization ability, as the tests include unseen combinations of visual concepts and rules, which is a classical intelligence quotient test.

Developing models that reveal human-like AVR ability remains a significant challenge in artificial intelligence research (Zheng et al., 2019; Chollet, 2019; Małkiński & Mańdziuk, 2022b). Many AVR benchmarks have been proposed to evaluate intelligent systems in a way that mirrors human-like AVR ability (Barrett et al., 2018; Hill et al., 2019; Zhang et al., 2019; Nie et al., 2020). These benchmarks usually adopt the problem structure of traditional psychological tests and provide programs to generate a large number of problems automatically, *e.g.*, the RAVEN dataset generates RPM problems by combining predefined visual concepts and abstract rules (Zhang et al., 2019). In recent years, deep AVR solvers have achieved remarkable progress on the proposed AVR benchmarks (Barrett et al., 2018; Zhuo & Kankanhalli, 2021; Mondal et al., 2023). However, they often rely on task-specific inductive biases of model architecture and hyperparameters, and solving different tasks typically means training different instances of models. This task-specific paradigm arouses interest in exploring a more unified framework for AVR solvers (Webb et al., 2024).

AVR tasks take different structures of problem panels to assess the subject's ability to understand abstract rules. RPM-style tasks (Barrett et al., 2018; Zhang et al., 2019) require discovering abstract rules of a $3 \times 3$ image matrix and selecting answers from a candidate panel to complete the matrix via analogy. Odd-one-out problems (McMillen, 2007) test the ability to identify the image that violates the abstract rules of a sequence. In addition, AVR tasks can be distinguished as generative tasks and selective tasks. Generative tasks require the synthesizing or reconstruction of images (Bar et al., 2022; Dedhia et al., 2023; Bai et al., 2023), while

[1]Shanghai Key Laboratory of Intelligent Information Processing, School of Computer Science, Fudan University. Correspondence to: Bin Li <libin@fudan.edu.cn>.

*Proceedings of the 42ⁿᵈ International Conference on Machine Learning*, Vancouver, Canada. PMLR 267, 2025. Copyright 2025 by the author(s).

selective tasks provide predefined options for the selection of answers (Barrett et al., 2018; Hill et al., 2019; Nie et al., 2020). These AVR tasks have different structures and goals, posing challenges in proposing a unified problem-solving framework.

Conditional generative models can discover the latent structure of observations, capture complex conditional dependencies between latent factors, and generate outputs conditioned on specific input variables or context (Mirza, 2014; Sohn et al., 2015; Chen et al., 2016). This ability is beneficial for AVR tasks that require understanding the abstract rules underlying the problem panels, where the latent structures can represent the abstract rules like shape transformations and spatial arrangements. Conditional generative models offer a powerful tool for solving AVR problems. Previous works have shown that both generative and selective AVR tasks can be tackled via conditional generation (Pekar et al., 2020; Shi et al., 2021; 2024). The solvers can generate the missing part of the panel and then compare it with the candidates to select the correct answer of an RPM test, which sheds light on unifying selective and generative AVR tasks.

We propose a novel framework, Unified Conditional Generative Solver (UCGS), to address multiple AVR tasks within a unified framework. UCGS frames various AVR tasks as a conditional generation process, which does not need repeated model training or task-specific hyperparameters when solving different AVR tasks. We prove that some classical AVR tasks, including *RPM* (Zhang et al., 2019; Barrett et al., 2018), *Visual Analogy Problem (VAP)* (Hill et al., 2019), *Odd-One-Out (O3)* (Mańdziuk & Żychowski, 2019) and *Synthetic Visual Reasoning Task (SVRT)* (Fleuret et al., 2011), can be solved by estimating the probability of generating the target images conditioned on the remaining images within a problem panel. We design a conditional generative network to instantiate the proposed framework, which can infer visual concepts from image patches and conduct abstract visual reasoning in terms of the concepts. The experiments demonstrate that UCGS exhibits not only the abstract reasoning ability on in-distribution tasks, but also the zero-shot reasoning ability to handle unseen tasks during the test phase.

## 2. Related Work

**Abstract Visual Reasoning.** Abstract visual reasoning (AVR) tasks are taken to measure the ability of abstract rule learning and problem-solving through analogical reasoning. Some early models (Lovett et al., 2010; Little et al., 2012) relied on artificially designed features to perform abstract visual reasoning. In recent years, a large number of models based on deep features have emerged (Barrett et al., 2018; Wang et al., 2020; Yun et al., 2020; Zhuo & Kankanhalli, 2021; Depeweg et al., 2024). Different abstract features,

such as hierarchical features (Zheng et al., 2019; Benny et al., 2021; Hu et al., 2021), disentangled representations (Van Steenkiste et al., 2019; Wu et al., 2020), and object-centric representations (Mondal et al., 2023; Webb et al., 2024), have been introduced as cues for abstract reasoning. Some approaches solve RPMs based on neuro-symbolic architectures (Hersche et al., 2023). On the other hand, some approaches focus on solving generative AVR tasks (Pekar et al., 2020; Shi et al., 2021; Zhang et al., 2021a;b; Shi et al., 2023; 2024). They directly complete problem panels from the given contexts and select answers by comparing the generated results with candidates. Inspired by language-prompted LLMs (Brown et al., 2020), visual in-context learners have been proposed to solve analogical reasoning problems in visual data (Bar et al., 2022; Dedhia et al., 2023; Bai et al., 2023; Zhao et al., 2023). Compared with the previous solvers, UCGS focuses on solving AVR tasks in a unified conditional generation framework.

**Deep Conditional Generative Models.** Deep generative models learn distributions of entire datasets, then they can generate novel samples that do not appear in the training stage (Kingma & Welling, 2013; Goodfellow et al., 2014; Du & Mordatch, 2019; Rombach et al., 2022). Deep conditional generative models (DCGMs) estimate the conditional probability distribution of the output given a certain input (Mirza, 2014; Sohn et al., 2015; Chen et al., 2016). Conditional Generative Adversarial Network (CGAN) (Mirza, 2014) is a classical type of DCGM, where a generator produces data conditioned on the input, and a discriminator distinguishes between the real and generated data. Conditional Variational Autoencoder (CVAE) (Sohn et al., 2015) introduces conditional inputs to conventional VAE (Kingma & Welling, 2013), where conditional inputs are encoded with input data into a latent space, and the latent representation is decoded to generate new samples in terms of the conditional inputs. Conditional generative models reveal a deep understanding of complex structures behind conditional inputs and data. Some DCGMs discover the underlying structure behind complex data by predicting the missing part of the data from the given context. Masked autoencoder (MAE) (He et al., 2022) is given an input where some portions are masked, and the task is to generate the missing pieces. The NP family (Garnelo et al., 2018; Kim et al., 2019; Dutordoir et al., 2023) can simulate a continuous function based on given points. In this paper, UCGS solves different AVR tasks with a unified conditional generative model, which extends the application of DCGMs in learning underlying structures from data.

## 3. Methods

In this section, we illustrate how to solve commonly used AVR tasks through conditional generation and leverage a

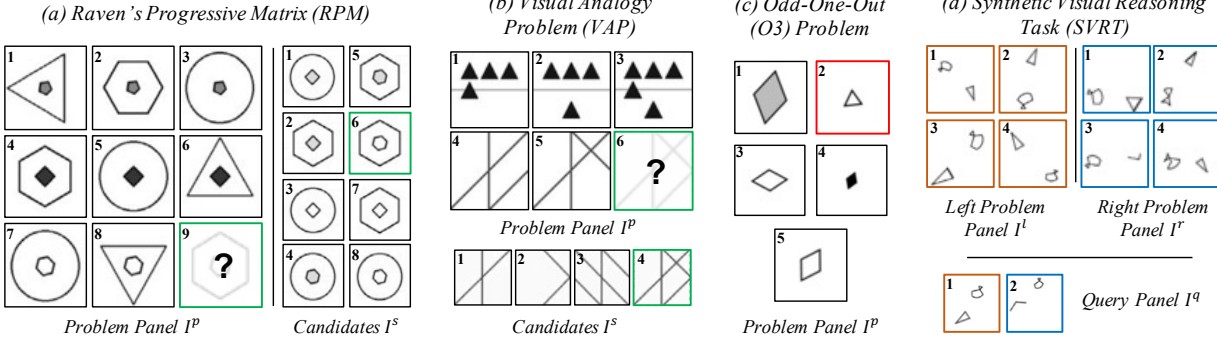

Figure 1: **An illustration of abstract visual reasoning (AVR) tasks.** This paper involves four AVR tasks, despite their differences in the predefined visual concepts (*e.g.*, object shape and color) and abstract rules (*e.g.*, progressive change and logical rule), all assess the ability to infer abstract rules from visual stimuli. *(a) Raven's Progressive Matrices (RPMs)* are visual puzzles where participants are given a problem panel with one missing piece. The task is to select the correct image (*e.g.*, the 6th candidate image in the example) from the candidate panel that completes the problem panel. *(b) Visual Analogy Problems (VAPs)* are similar to RPMs. The participant must choose candidate images that complete the analogy and fit the missing piece of problem panels. *(c)* In an *Odd-One-Out (O3) problem*, solvers are given a problem panel to find the odd image that breaks the abstract rule (*e.g.*, the 2nd image in the example panel). *(d)* A problem of *Synthetic Visual Reasoning Task (SVRT)* consists of two problem panels following different abstract rules. The task is to deduce the rule that differentiates the problem panels, and solvers are required to categorize the query image into the left or right panel.

unified conditional generative network for abstract visual reasoning. We then introduce an instance of UCGS, where the conditional generative network can generate missing panels from context across various tasks.

### 3.1. Unified Conditional Generative Solver

As shown in Figure 1, the AVR tasks define different visual concepts and the abstract rules on the concepts, which test the understanding of abstract rules in different ways of obtaining answers. We will introduce how to solve different AVR tasks in a unified way as follows.

**Definition 3.1.** $I^p = \{I_i^p | i = 1, \ldots, N\}$ is a problem panel where $I_i^p$ is the $i$-th panel image of $I^p$. The *correctness* of $I^p$ is the joint probability $p(I^p)$. $I^p$ is *rule-compliant* if $p(I^p)$ is large, and *rule-violating* if $p(I^p) \to 0$.

By modeling the correctness of a problem panel as a joint probability over panel images, we can define the conditional probability of predicting a specific image in the panel.

**Definition 3.2.** For panel $I^p$, the *predictability* $p(x|I_{\neg i}^p)$ indicates the probability that the $i$-th panel image is $x$ given the context $I_{\neg i}^p$, where $I_{\neg i}^p = \{I_j^p\}_{j \neq i}$.

Estimating and sampling results from the predictability can solve generative RPM problems, since the problems require solvers to generate answers from the context (Pekar et al., 2020). But the predictability cannot directly solve selective RPM problems where the solvers must pick answers from

candidate panels (Zhang et al., 2019). Let $I_{i \to x}^p$ be the panel created by replacing the $i$-th image of the panel $I^p$ to $x$. Using Definitions 3.1 and 3.2, UCGS can solve the following selective tasks by estimating the predictability.

**Proposition 3.3.** *Given the problem panel $I^p$ and candidate panel $I^s$ of an RPM test or VAP, where $N = |I^p|$, the correct answer $x^\star = \text{argmax}_{x \in I^s} p(x|I_{\neg N}^p)$.*

*Proof.* The context panel $I_{\neg N}^p$ is created by removing the last image from $I^p$. The modified panels $\{I_{N \to x}^p | x \in I^s\}$ are created by placing the candidate images to the removed position. The correctness of $I_{N \to x}^p$ is

$$p(I_{N \to x}^p) = p(x|I_{\neg N}^p)p(I_{\neg N}^p) \propto p(x|I_{\neg N}^p). \quad (1)$$

The correct answer $x^\star$ make up the modified panel with the largest correctness, given by $\text{argmax}_{x \in I^s} p(x|I_{\neg N}^p)$. $\square$

**Proposition 3.4.** *Given the panel $I^p$ of an O3 problem, the index of the odd image $i^\star = \text{argmin}_{i=1,\cdots,N} p(I_i^p|I_{\neg i}^p)$, where $N = |I^p|$.*

*Proof.* We create context panels $\{I_{\neg i}^p | i = 1, \cdots, N\}$ by removing each image from $I^p$. The correctness of $I_{\neg i}^p$ is

$$p(I_{\neg i}^p) = \frac{p(I^p)}{p(I_i^p|I_{\neg i}^p)} \propto \frac{1}{p(I_i^p|I_{\neg i}^p)}. \quad (2)$$

If $I_i^p$ is the odd image, $p(I_{\neg i}^p)$ will be large since the rule-breaking image is removed, otherwise $I_{\neg i}^p$ is still rule-

violating and $p(\boldsymbol{I}^p_{\neg i}) \to 0$. Hence the index of the odd image is $i^\star = \arg\min_{i=1,\cdots,N} p(\boldsymbol{I}^p_i|\boldsymbol{I}^p_{\neg i})$. $\qquad\square$

**Proposition 3.5.** *Given a left problem panel $\boldsymbol{I}^l$, a right problem panel $\boldsymbol{I}^r$ and a query panel $\boldsymbol{I}^q$ of SVRT, where $N = |\boldsymbol{I}^l| = |\boldsymbol{I}^r|$, the query image $\boldsymbol{x} \in \boldsymbol{I}^q$ belongs to the panel $\boldsymbol{I}^\star = \arg\max_{\boldsymbol{I}^p \in \{\boldsymbol{I}^l, \boldsymbol{I}^r\}} p(\boldsymbol{x}|\boldsymbol{I}^p)$.*

*Proof.* By appending the query image $\boldsymbol{x} \in \boldsymbol{I}^q$ to the end of the problem panels $\boldsymbol{I}^l$ and $\boldsymbol{I}^r$, we get the extended panels $\boldsymbol{I}^l_{+\boldsymbol{x}}$ and $\boldsymbol{I}^r_{+\boldsymbol{x}}$ . The correctness of the extended panels is

$$
\begin{aligned}
p(\boldsymbol{I}^l_{+\boldsymbol{x}}) &= p(\boldsymbol{x}|\boldsymbol{I}^l)p(\boldsymbol{I}^l), \\
p(\boldsymbol{I}^r_{+\boldsymbol{x}}) &= p(\boldsymbol{x}|\boldsymbol{I}^r)p(\boldsymbol{I}^r).
\end{aligned}
\tag{3}
$$

If $\boldsymbol{x}$ belongs to $\boldsymbol{I}^p$, the extended panel $\boldsymbol{I}^p_{+\boldsymbol{x}}$ remains rule-compliant because the panel rule is not changed. Therefore, $\boldsymbol{I}^p_{+\boldsymbol{x}}$ has the largest correctness. Consider that $\boldsymbol{I}^l$ and $\boldsymbol{I}^r$ are sampled from the dataset $\mathcal{D}$ uniformly without overlap, in Equation 3 we have $p(\boldsymbol{I}^l) = p(\boldsymbol{I}^r) = 1/|\mathcal{D}|$. Therefore, $\boldsymbol{x}$ belongs to the panel $\boldsymbol{I}^\star = \arg\max_{\boldsymbol{I}^p \in \{\boldsymbol{I}^l, \boldsymbol{I}^r\}} p(\boldsymbol{x}|\boldsymbol{I}^p)$. $\square$

Propositions 3.3, 3.4, and 3.5 demonstrate that although the answers to AVR problems have different forms, they can be converted into the estimation of predictability on panel images. In this way, answer selection and generation are unified. If the $i$-th image of an RPM problem panel is missing, we can generate answers by sampling from $p(\boldsymbol{x}|\boldsymbol{I}^p_{\neg i})$, as well as select answers by filling in the candidate $\boldsymbol{x}$ and computing the panel correctness $p(\boldsymbol{I}^p_{i \to \boldsymbol{x}}) = p(\boldsymbol{x}|\boldsymbol{I}^p_{\neg i})p(\boldsymbol{I}^p_{\neg i})$. With a unified predictability estimator, UCGS can solve different AVR problems after a round of multi-task training, preventing repeated training or tuning of AVR solvers. The core of UCGS is a shared predictability estimator and training-free judgment functions. The predictability estimator is a conditional generative network trained on different AVR tasks to estimate the predictability of panel images. As shown in Propositions 3.3, 3.4, and 3.5, the judgment functions transform outputs of the predictability estimator to obtain the final results, which are task-specific. In the next sections, we provide an instance of UCGS and introduce the architecture of the conditional generative network in detail.

## 3.2. Instantiation of UCGS

We instantiate UCGS with a Transformer-based conditional generative network, called UCGS-T. Figure 2a illustrates the architecture of UCGS-T via an RPM problem. The problem panel of an RPM contains $N = 9$ images, where we leave one as the prediction target and the others as the context. We denote the index of the target image as $t$, and the indices of the context images as a set $C$. The objective of UCGS-T is to estimate the predictability $p(\boldsymbol{I}^p_t|\boldsymbol{I}^p_C)$. UCGS-T consists of five modules, which will be introduced as follows.

### 3.2.1. IMAGE ENCODER AND DECODER

Figure 2b depicts the architecture of the image encoder and decoder. UCGS-T learns patch representations of the input image by mapping continuous representations into a finite set of discrete codes, which is then decoded back into the original data space (Van Den Oord et al., 2017; Razavi et al., 2019). The image encoder extracts a feature map with $M$ visual features from $\boldsymbol{I}^p_i$ using a CNN-based neural network, which are projected to the nearest vectors in a fixed-size codebook $\boldsymbol{e} = (\boldsymbol{e}_1, \boldsymbol{e}_2, \ldots, \boldsymbol{e}_L)$ to obtain $M$ quantized patch representations $\boldsymbol{Z}_i = \{\boldsymbol{Z}_{i,1}, \boldsymbol{Z}_{i,2}, \ldots, \boldsymbol{Z}_{i,M}\}$. The image decoder takes the quantized patch representations as input and reconstructs the image from the discrete representation space. The input images are compressed into quantized patch representations to learn more abstract and general representations for reasoning. We can estimate the predictability $p(\boldsymbol{I}^p_t|\boldsymbol{I}^p_C)$ over the high-dimensional images through $p(\boldsymbol{Z}_t|\boldsymbol{Z}_C)$ on the patch representations to reduce the complexity of reasoning.

### 3.2.2. PATCH ENCODER

The quantized patch representations focus on local regions of the image, potentially corresponding to a specific geometric shape or pattern learned by the codebook. As shown in Figure 2c, the patch encoder computes the relation between the local patch representations to understand the role of each patch in the overall image and extract the image-level visual concepts (*e.g.*, the number of entities in the image). We add $M$ learnable positional embeddings to the patches of each panel image, which is projected to the high-dimensional representations $\tilde{\boldsymbol{Z}}$ by a linear network to encode spatial position information, where $\tilde{\boldsymbol{Z}}_{i,m}$ is the position-augmented representations of $\boldsymbol{Z}_{i,m}$ for $i = 1, \ldots, N$ and $m = 1, \ldots, M$. $\tilde{\boldsymbol{Z}}_i$ are passed through a Transformer decoder to capture global dependencies and relationships between the patches. In addition to the patch representations, we introduce class tokens $\{\mathrm{CLS}_1, \ldots, \mathrm{CLS}_K\}$ as learnable slots to aggregate global information from the patches and obtain the visual concepts $\boldsymbol{S}_i = \{\boldsymbol{S}_{i,k}|k = 1, \ldots, K\}$ where

$$
\boldsymbol{S}_{i,k} = \mathrm{TransformerDecoder}(\mathrm{CLS}_k, \tilde{\boldsymbol{Z}}_i).
\tag{4}
$$

The independent slots capture $K$ visual concepts from the patches. The Transformer decoder updates the slots based on attention mechanisms, where the slots act as queries and the patch representations $\tilde{\boldsymbol{Z}}_i$ are keys and values. The final output of the patch encoder is the visual concepts $\boldsymbol{S}_i$ that consider relationships across the patches of the entire image.

### 3.2.3. CONCEPT ENCODER

The concept encoder combines the visual concepts of the context images (*i.e.*, context concepts) $\boldsymbol{S}_C = \{\boldsymbol{S}_i|i \in C\}$, understanding the overall layout and abstract rule within the

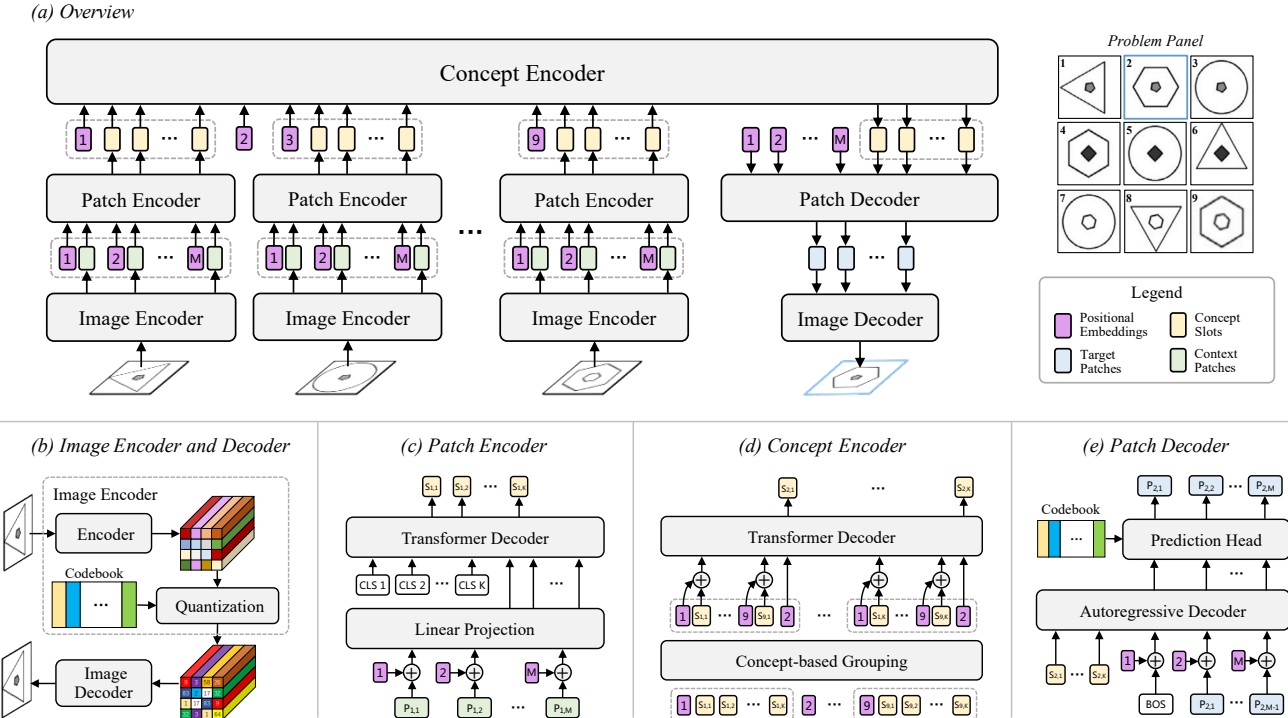

Figure 2: **An Overview of UCGS-T.** The image encoder extracts high-level features from each context panel image, which are mapped into discrete context patch representations via vector quantization. The patch encoder captures image-level visual concepts from context patches that encode local information. The panel encoder integrates visual concepts of the context images, understanding abstract rules on the panel, and predicting visual concepts for the target image. The patch decoder generates patch representations of the target image from the target visual concepts predicted by the panel encoder. The image decoder reconstructs the target image from the target patches.

panel, and predicting the visual concepts of the target image (*i.e.*, target concepts) $\boldsymbol{S}_t$. Figure 2d displays the architecture of the concept encoder. First, the input context concepts $\boldsymbol{S}_C$ are reorganized into $K$ groups $\{\boldsymbol{G}_1, \ldots, \boldsymbol{G}_K\}$ in terms of the concepts, where the $k$-th group $\boldsymbol{G}_k = \{\boldsymbol{S}_{i,k} | i \in C\}$. We group the context concepts since the abstract rules are typically global and shared among different visual concepts (Wu et al., 2020; Shi et al., 2024), *e.g.*, the tuples *(Triangle, Square, Pentagon)* and *(Small, Middle, Large)* point to the same abstract rule *Increase*. To encode the position of concepts within the group $\boldsymbol{G}_k$, learnable positional embeddings are added to the visual concepts in $\boldsymbol{G}_k$, followed by a linear projection to produce the position-augmented representations $\tilde{\boldsymbol{G}}_k$. Each group is processed independently by a shared Transformer decoder to capture concept-specific abstract rules and predict the target visual concepts $\boldsymbol{S}_t = \{\boldsymbol{S}_{t,k} | k = 1, \ldots, K\}$ where

$$\boldsymbol{S}_{t,k} = \text{TransformerDecoder}(\text{PE}_t, \tilde{\boldsymbol{G}}_k). \quad (5)$$

$\text{PE}_t$ is a learnable target positional embedding. The output $\boldsymbol{S}_t$ is used to reconstruct the patches of the target image.

### 3.2.4. PATCH DECODER

The patch decoder in Figure 2e generates target patches from the target visual concepts $\boldsymbol{S}_t$ in an autoregressive manner (Yan et al., 2021). The predictability is factorized by

$$p(\boldsymbol{Z}_t | \boldsymbol{Z}_C) = \prod_{m=1}^{M} p(\boldsymbol{Z}_{t,m} | \boldsymbol{Z}_{t,<m}, \boldsymbol{Z}_C) \quad (6)$$

where a learnable token $\boldsymbol{Z}_{t,0}$ is used to indicate the beginning of the autoregressive decoding process. The process generates the patch sequence one step at a time, where the $m$-th step depends on the previously generated patches $\boldsymbol{Z}_{t,<m}$ and the target concepts $\boldsymbol{S}_t$ computed from $\boldsymbol{Z}_C$. The $m$-th step $p(\boldsymbol{Z}_{t,m} | \boldsymbol{Z}_{t,<m}, \boldsymbol{Z}_C)$ is parameterized by a Transformer decoder with a prediction head:

$$
\begin{aligned}
\tilde{\boldsymbol{Z}}_{t,<m} &= \text{PostionalEmbedding}\left(\boldsymbol{Z}_{t,<m}\right), \\
\boldsymbol{H} &= \text{TransformerDecoder}\left(\tilde{\boldsymbol{Z}}_{t,<m}, \boldsymbol{S}_t\right), \\
\boldsymbol{\pi}_m &= \text{Softmax}\left(\text{Linear}\left(\boldsymbol{H}_m\right)\right), \\
\boldsymbol{Z}_{t,m} &= \boldsymbol{e}\left[c_m\right], \text{ where } c_m \sim \text{Categorical}\left(\boldsymbol{\pi}_m\right).
\end{aligned}
\quad (7)
$$

The output of the Transformer decoder is passed through a prediction head, which maps the hidden representations into the probabilities over the discrete codes in the codebook. The final prediction of the target patch $\boldsymbol{Z}_{t,m}$ is given by querying the $c_m$-th vector $\boldsymbol{e}[c_m]$ in the codebook.

### 3.2.5. MODEL TRAINING

The total loss of the model consists of an image reconstruction loss $\mathcal{L}_{\text{recon}}$ and a patch prediction loss $\mathcal{L}_{\text{pred}}$. We adopt the training objective of VQVAE (Van Den Oord et al., 2017) as the image reconstruction loss to ensure that the input image can be correctly reconstructed from its quantized discrete representations. $\mathcal{L}_{\text{recon}}$ is defined as the distance between the input images and the reconstructions with a commitment loss to ensure the continuous representations are not too far from the codebook vectors. The patch prediction loss evaluates the difference between the predicted target patches (discrete codes) and the target patches encoded from the image encoder. $\mathcal{L}_{\text{pred}}$ is defined as

$$\mathcal{L}_{\text{pred}} = -\sum_{m=1}^{M} \log p(\boldsymbol{Z}_{t,m}|\boldsymbol{Z}_{t,<m}, \boldsymbol{Z}_C). \qquad (8)$$

By combining the image reconstruction loss and patch prediction loss, the total loss is given as

$$\mathcal{L}_{\text{total}} = \mathcal{L}_{\text{pred}} + \lambda \cdot \mathcal{L}_{\text{recon}} \qquad (9)$$

where $\lambda$ is a hyperparameter that balances the losses. The image encoder and decoder are pretrained before training the remaining modules. We can set $\lambda > 0$ to finetune the image encoder and decoder in the following training stage. But we find that freezing their parameters is the best choice.

## 4. Experiments

In this section, we compare UCGS-T with ablation baselines, task-specific solvers and Multimodal LLMs (MLLMs) on four classical AVR tasks shown in Figure 1. We also conduct qualitative experiments to illustrate the performance of UCGS-T in answer generation tasks by visualizing the generation results of RAVEN and PGM.

**Datasets.** The models are evaluated on RAVEN (Zhang et al., 2019) and PGM (Barrett et al., 2018) to assess their reasoning abilities on RPM tasks. The ability to reason over O3 problems is tested using the G1-set dataset (Mańdziuk & Żychowski, 2019). Performance on the remaining AVR tasks is evaluated using the VAP (Hill et al., 2019) and SVRT (Fleuret et al., 2011) datasets. RAVEN and PGM are used for both training and testing, while G1-set, VAP, and SVRT are used only during testing to assess the zero-shot reasoning capabilities of the models. In addition to the public datasets, we construct three new datasets, *i.e.*, O3-ID, VAP-ID, and SVRT-ID, based on RAVEN. These

datasets are designed to evaluate the ability to reason on unseen AVR tasks that share in-distribution visual concepts and abstract rules with the training data. See Appendix A for more details on the dataset construction.

**Metric.** We compute the selection accuracy, a standard metric that evaluates model performance on AVR tasks (Barrett et al., 2018). This metric is applicable to both selective and generative solvers. For selective solvers, which predict the index of answers, selection accuracy can be calculated straightforwardly. Although candidate panels guarantee a unique correct answer, many AVR problems admit multiple valid solutions, making it challenging to enumerate all possible correct answers when evaluating generative solvers (Shi et al., 2024). We consider the output of a generative solver to be correct if it is the closest match to the ground truth answer among the candidate panels. Accordingly, both selective and generative solvers are evaluated using selection accuracy in our experiments.

**Ablation Baselines.** We employ three backbone architectures and two conditional generative solvers to construct a set of ablation baselines. One of the backbones is the patch-based backbone (Patch) used in UCGS-T. Given that recent task-specific solvers (Mondal et al., 2023; Webb et al., 2024) have demonstrated strong performance using object-centric representations, we introduce an object-centric backbone (OCL) (Locatello et al., 2020; Yuan et al., 2023), which extracts slot representations to capture individual objects in panel images. We also incorporate a monolithic backbone (Mono) that encodes each panel image as a single representation. We introduce two baseline conditional generative solvers for predictability estimation. The Transformer-based solver (TF) maps context images to target images using a Transformer encoder-decoder architecture. The ANP-based solver (ANP) (Kim et al., 2019) models the distribution over image panels using stochastic functions where context images are used to infer the functions, and target images are sampled from fixed locations of the functions. By combining the backbones and conditional generative solvers, we construct six ablation baselines. Further details of UCGS-T and these baselines are provided in Appendix B. UCGS-T and the ablation baselines are trained under a multi-task setting using all seven configurations of RAVEN and the neutral configuration of PGM. The models are evaluated on VAP, G1-set, and SVRT without retraining or fine-tuning to assess zero-shot reasoning ability. We conduct qualitative experiments by visualizing the target images generated by the models to illustrate and compare the answer-generation capabilities of UCGS-T and the ablation baselines.

**Task-specific Solvers and MLLMs.** Since UCGS-T is designed to handle both selective and generative tasks, we compare it against several task-specific solvers that support both answer selection and generation: PrAE (Zhang

Table 1: **Selection accuracy (%) on AVR tasks.** ID tasks have only in-distribution visual concepts and abstract rules, while OOD tasks contain out-of-distribution visual concepts and abstract rules. The problems of zero-shot tasks only appear in the testing stage. The models marked with * are evaluated on subsets of the datasets.

| | ID Tasks | | ID and Zero-Shot Tasks | | | OOD and Zero-Shot Tasks | | |
| Model | RAVEN | PGM | O3-ID | VAP-ID | SVRT-ID | G1-set | VAP | SVRT |
|---|---|---|---|---|---|---|---|---|
| ANP + Mono | 6.0 | 14.1 | 11.9 | 5.8 | 51.9 | 32.9 | 30.7 | 48.4 |
| ANP + OCL | 8.3 | 12.3 | 13.3 | 8.5 | 49.4 | 29.3 | 30.4 | 49.8 |
| ANP + Patch | 10.5 | 12.3 | 11.7 | 10.9 | 49.8 | 30.6 | **30.9** | 51.0 |
| TF + Mono | 11.2 | 14.1 | 13.6 | 6.5 | 56.4 | 32.7 | 30.2 | 50.8 |
| TF + OCL | 7.4 | 12.3 | 12.7 | 7.1 | 49.6 | **33.7** | 30.6 | 50.0 |
| TF + Patch | 21.0 | 15.6 | **35.8** | 15.5 | 71.8 | 30.5 | 27.1 | 51.4 |
| UCGS-T | **64.6** | **38.1** | 33.6 | **35.8** | **84.6** | 30.4 | 28.8 | **52.8** |
| GPT-4o* | 12.6 | 21.4 | 30.4 | 12.6 | 40.8 | **52.9** | 29.3 | **64.2** |
| UCGS-T* | **60.9** | **37.8** | **32.7** | **34.0** | **85.9** | 30.4 | **30.1** | 52.8 |
| Random Guess | 12.5 | 12.5 | 11.1 | 12.5 | 50.0 | 22.5 | 25.0 | 50.0 |

et al., 2021a), NVSA (Hersche et al., 2023), GCA (Pekar et al., 2020), ALANS (Zhang et al., 2021b), and RAISE (Shi et al., 2024). To ensure consistency with the experimental setup used for UCGS-T and the ablation baselines, all task-specific solvers are trained and tested on each dataset separately, without supervision from rule or attribute annotations. In addition, we include GPT-4o (Achiam et al., 2023), a powerful general-purpose multimodal language model (MLLM), as a reference point for comparison. Task-specific prompts for GPT-4o are designed based on the prior benchmark study (Cao et al., 2024).

### 4.1. Performance on AVR Tasks

The experiments evaluate model performance across four AVR tasks. We categorize the evaluation settings into the following three groups. *In-Distribution (ID) Tasks* include problems that are seen during training, whose abstract rules and visual concepts are also encountered during training. *In-Distribution Zero-Shot (ID-ZS) Tasks* are not used during training, but their abstract rules and visual concepts are present in the training data. *Out-of-Distribution Zero-Shot (OOD-ZS) Tasks* are the most challenging tasks, which are entirely unseen during training and involve novel abstract rules or visual concepts. We compare UCGS-T with both ANP-based ablation baselines (ANP + Mono, ANP + OCL, ANP + Patch) and Transformer-based baselines (TF + Mono, TF + OCL, TF + Patch), as well as GPT-4o. For reference, we also include a random guess baseline.

#### 4.1.1. In-Distribution Tasks

UCGS-T and the baselines are trained on RAVEN and PGM, whose test splits are treated as in-distribution (ID) tasks. While the training and testing samples of RAVEN and PGM

are generated using the same sets of abstract rules and visual concepts, the rules and concepts are composed differently between the splits. The purpose of the ID tasks is to evaluate the ability to reason over the abstract rules and visual concepts that have been learned during training. As shown in Table 1, Transformer-based baselines outperform their ANP-based counterparts, with TF + Patch achieving the strongest results among them. UCGS-T further achieves 64.6% accuracy on RAVEN and 38.1% on PGM, demonstrating the abstract reasoning capabilities on ID tasks. In contrast, ANP-based baselines perform poorly, with accuracies close to random guessing. This suggests that ANP-based models struggle with compositional reasoning in AVR tasks such as RAVEN and PGM.

#### 4.1.2. In-Distribution Zero-Shot Tasks

We introduce ID-ZS tasks to evaluate the ability of models to generalize their reasoning capabilities to novel tasks. In this setting, all models are trained on RPM tasks and evaluated on other types of AVR tasks. They are required to solve problems from zero-shot tasks that contain in-distribution abstract rules and visual concepts. We construct three ID-ZS datasets, *i.e.*, O3-ID, VAP-ID, and SVRT-ID, which follow the forms of O3, VAP, and SVRT, respectively. These datasets are built on the abstract rules and visual concepts of RAVEN. Across all ID-ZS tasks, models exhibit a noticeable drop in performance compared to ID tasks, highlighting the difficulty of transferring learned reasoning abilities to new task formats. Transformer-based baselines continue to outperform their ANP-based counterparts. UCGS-T achieves the best results overall, with accuracies of 33.6% on O3-ID, 35.8% on VAP-ID, and 84.6% on SVRT-ID. These results suggest that UCGS-T is more effective at generalizing rea-

soning abilities from the training data to novel tasks. For example, UCGS-T can perform abstract reasoning on VAPs with six-image panels, despite having been trained solely on RPMs with nine-image panels.

### 4.1.3. OUT-OF-DISTRIBUTION ZERO-SHOT TASKS

OOD-ZS tasks evaluate models on novel abstract rules and visual concepts that are entirely unseen during training, with test problems presented in forms that differ from those used in the training phase. We use the G1-set, VAP, and SVRT datasets to assess the out-of-distribution and zero-shot reasoning capabilities of models. As shown in Table 1, both UCGS-T and the baselines experience a substantial drop in performance compared to the ID tasks. On G1-set and VAP, UCGS-T and the baselines achieve accuracies around 30%, which is 5–8% higher than random guessing. This indicates that the UCGS framework enables a certain degree of generalization in abstract reasoning. UCGS-T slightly outperforms the other baselines on SVRT, a task that requires holistic pattern recognition based on spatial relationships among randomly generated abstract shapes. This requirement is similar to image-level visual concept recognition in datasets such as PGM and RAVEN (*e.g.*, recognizing the number of objects in a scene). The results on SVRT suggest that UCGS-T may benefit from its panel encoder, which is specifically designed to extract high-level image abstractions. Furthermore, UCGS-T achieves a large accuracy improvement over the baselines in ID tasks, while the performance gains are relatively smaller in OOD tasks. We suppose that the unseen visual concepts and abstract rules in OOD tasks lead to a significant drop in accuracy for both UCGS-T and the baselines, thereby narrowing the performance gap that arises from different model architectures.

### 4.1.4. COMPARISON WITH GPT-4O

Table 1 also presents the results of the general-purpose system GPT-4o. We randomly sampled subsets from the test sets of RAVEN, PGM, and VAP to evaluate both GPT-4o and UCGS-T. The results show that UCGS-T outperforms GPT-4o significantly on ID and ID-ZS tasks, while GPT-4o achieves better performance on OOD-ZS tasks. This may be attributed to GPT-4o's ability to recognize unseen visual concepts or abstract rules. However, this does not necessarily imply that GPT-4o possesses stronger zero-shot reasoning ability, as it is difficult to determine whether similar visual concepts or rules were encountered during its training. In contrast, UCGS-T is trained on a clearly defined and controlled set of samples, making its evaluation setting more transparent. Notably, GPT-4o shows weaker generalization on ID-ZS tasks, suggesting that UCGS-T is more effective at transferring its reasoning capabilities to novel task formats. In this experiment, GPT-4o is provided with task-specific prompts, following the setup in (Cao et al., 2024). We also

Table 2: **Selection accuracy (%) of UCGS-T and the task-specific solvers on RAVEN and PGM.** The task-specific solvers are trained and tested separately on each dataset. They are trained without additional annotations to keep consistent with the experimental setup of UCGS-T and baselines. PrAE and ALANS only define the architecture to solve RAVEN. The codebase of NVSA is not applicable to PGM, and the reported accuracy on PGM is obtained by using additional annotation information. We only report their performance on RAVEN here.

| MODEL | RAVEN | PGM |
|---|---|---|
| PRAE (ZHANG ET AL., 2021A) | 13.6 | - |
| NVSA (HERSCHE ET AL., 2023) | 11.5 | - |
| GCA (PEKAR ET AL., 2020) | 37.3 | 31.7 |
| ALANS (ZHANG ET AL., 2021B) | 50.1 | - |
| RAISE (SHI ET AL., 2024) | 54.5 | 14.0 |
| UCGS-T | **64.6** | **38.1** |
| RANDOM GUESS | 12.5 | 12.5 |

incorporate prior knowledge in the prompts, such as stating that an RPM problem is a 3 × 3 matrix, which is not provided to UCGS-T during either training or testing. Prior work (Cao et al., 2024) has shown that GPT-4o achieves a notable performance boost (approximately 20%) when provided with language descriptions of candidate images. Although GPT-4o is a powerful general-purpose model, it is not specifically designed for AVR tasks. Its performance on RPM tasks may benefit from more advanced prompt engineering and chain-of-thoughts design. Appendix B.4 provides the prompts used in this experiment.

### 4.1.5. COMPARISON WITH TASK-SPECIFIC SOLVERS

This experiment compares UCGS-T with classic generative task-specific RPM solvers. Table 2 shows the experimental results where UCGS-T outperforms the task-specific solvers under the same experimental setup. The experimental results reveal that, without additional annotation information, the performance of generative task-specific solvers drops significantly. For example, PrAE achieves an accuracy of 65.0% with rule supervision during training (Zhang et al., 2021a), but its accuracy drops to 13.6% in our experiment. The reasoning process of GCA is specifically designed for predicting the bottom-right image in an RPM matrix, which limits its ability to generalize to other AVR tasks. Additionally, models like PrAE, ALANS, and NVSA rely on predefined rules and explicit representations tailored to specific datasets, making it difficult for them to handle unseen or undefined visual concepts and abstract rules. UCGS-T does not rely on manually designed concepts or rules. It performs an independent reasoning process over each visual concept, inferring both the underlying visual concepts and abstract rules without the need for additional annotations.

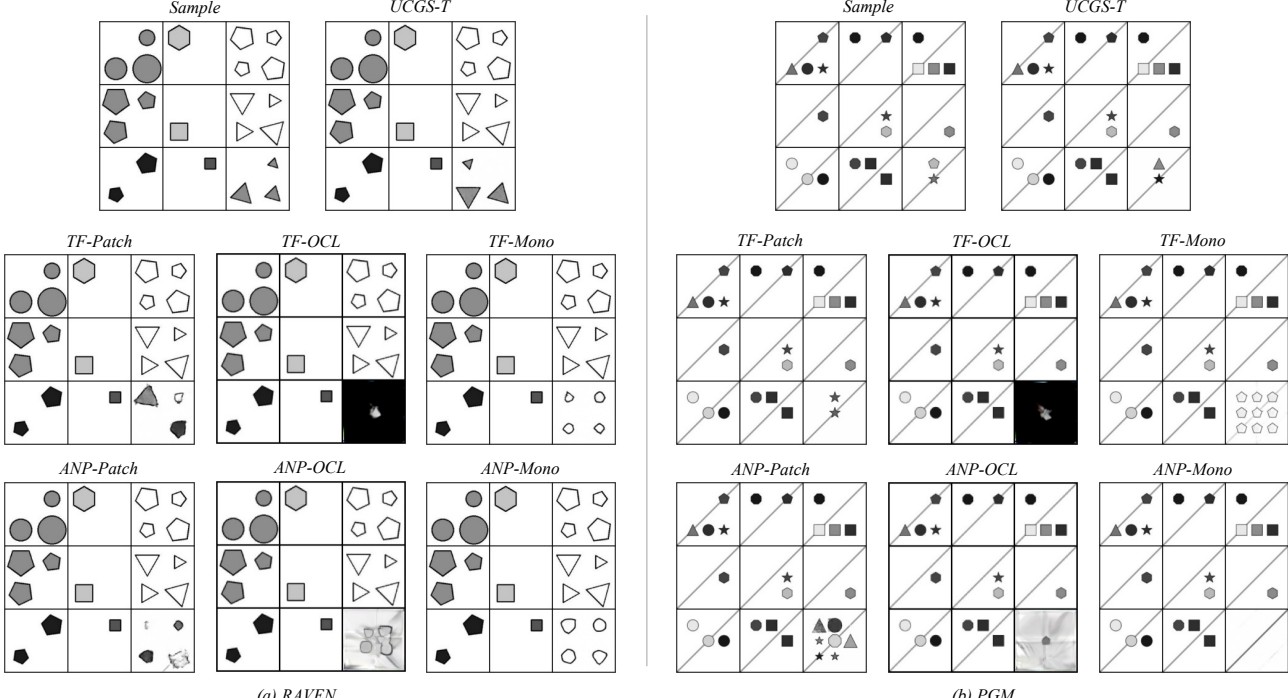

Figure 3: **Comparison of generation results on RAVEN (left) and PGM (right).** In the visualization result of each model, the bottom-right position is the prediction and the remaining context images are ground truths. The figure illustrates qualitative differences between models, with errors and artifacts appearing in some results predicted by baselines. Note that there is noise in the problem panel, therefore the generated result of UCGS-T on PGM is correct though it is a little different from the ground truth.

## 4.2. Answer Generation on RAVEN and PGM

Figure 3 visualizes the generated answers produced by UCGS-T and the baselines. UCGS-T can generate rule-compliant answers from the context on RAVEN and PGM. While TF-Patch is capable of generating clear and reasonable outputs on the PGM example, it fails to produce the correct answer for the RAVEN problem. Other baselines often generate images containing incorrect visual concepts. For instance, the solutions predicted by TF-Mono and ANP-Mono on RAVEN display scenes with four objects, whereas the correct answer should contain three objects. Models using the object-centric backbone generally struggle to generate high-quality outputs where the mismatch between predicted target slots and the true scene structure significantly degrades the pixel reconstruction quality. We observe that UCGS-T and TF-Patch are better at capturing diversity in answer generation. Since RAVEN and PGM introduce a degree of randomness or noise in data generation, UCGS-T's outputs may differ slightly from the ground truth but still conform to the task rules. For example, in Figure 3a, the answer should contain three objects, but their positions within the panel may vary without affecting correctness.

## 5. Conclusion and Limitations

We propose UCGS, a unified AVR framework to solve multiple AVR tasks using a single conditional generative network. We instantiate UCGS with a model that infers visual concepts from image patches to perform reasoning on the problem panel. Experimental results demonstrate that UCGS not only exhibits abstract reasoning ability on ID tasks but also zero-shot generalization, enabling it to solve unseen tasks during testing, even when those tasks involve OOD visual concepts and abstract rules.

**Limitations.** While UCGS-T outperforms the generative task-specific RPM solvers, its accuracy is still lower than state-of-the-art selective solvers. Reducing the performance gap remains an important direction for future work. We believe this paper provides a unified perspective for solving AVR tasks. This work does not address AVR tasks involving real-world scenes. Real-world problems introduce more complex visual concepts and abstract rules, which may require more powerful image tokenizers and decoders. Effectively discovering and utilizing such rules in realistic settings remains a challenge for future investigation.

## Impact Statement

This paper presents work whose goal is to construct unified abstract visual reasoning solvers. We feel that there are no potential societal impacts that must be specifically highlighted and discussed here.

## Acknowledgements

This work was supported by the National Natural Science Foundation of China (No.62176060) and the Program for Professor of Special Appointment (Eastern Scholar) at Shanghai Institutions of Higher Learning.

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

## A. Datasets

In this section, we will introduce the datasets used for model training and testing. Tables 3 and 4 introduce the training, validation, and test splits of the datasets.

Table 3: **The ID tasks used to train and evaluate the baselines, task-specific solvers and UCGS-T.** #Samples = number of samples per dataset, #Images = number of images per problem, #Candidates = number of candidate per problem.

| Dataset | RAVEN | | | PGM | | |
|---|---|---|---|---|---|---|
| Split | Train | Valid | Test | Train | Valid | Test |
| #Samples | 42K | 14K | 14K | 1.4M | 5K | 200K |
| #Images | 9 | | | 9 | | |
| #Candidates | 8 | | | 8 | | |
| Image Size | 128×128 | | | 128×128 | | |

Table 4: **The ID-ZS and OOD-ZS tasks used to evaluate the baselines and UCGS-T.** #Samples = number of samples per dataset, #Images = number of images per problem, #Candidates = number of candidate per problem.

| Dataset | G1-set | VAP | SVRT | O3-ID | VAP-ID | SVRT-ID |
|---|---|---|---|---|---|---|
| Split | Test | Test | Test | Test | Test | Test |
| #Samples | 1K | 200K | 253 | 14K | 14K | 14K |
| #Images | 4∼5 | 6 | 8 | 5 | 6 | 8 |
| #Candidates | NA | 4 | 2 | NA | 8 | 2 |
| Image Size | 128×128 | 128×128 | 128×128 | 128×128 | 128×128 | 128×128 |

Table 5: **The subsets used to evaluate GPT-4o and UCGS-T.** The datasets noted with * are subsets sampled from the corresponding complete datasets. #Samples = number of samples per dataset, #Images = number of images per problem, #Candidates = number of candidate per problem.

| Dataset | RAVEN* | PGM* | G1-set | VAP* | SVRT | O3-ID | VAP-ID | SVRT-ID |
|---|---|---|---|---|---|---|---|---|
| Split | Test | Test | Test | Test | Test | Test | Test | Test |
| #Samples | 700 | 1K | 1K | 1K | 253 | 700 | 700 | 700 |
| #Images | 9 | 9 | 4∼5 | 6 | 8 | 5 | 6 | 8 |
| #Candidates | 8 | 8 | NA | 4 | 2 | NA | 8 | 2 |
| Image Size | 128×128 | 128×128 | 128×128 | 128×128 | 128×128 | 128×128 | 128×128 | 128×128 |

### A.1. ID Tasks

**RAVEN** (Zhang et al., 2019). The RAVEN dataset is inspired by the widely used RPM tests. RAVEN provides a structured and systematically generated set of problems with clearly defined rules and visual concepts. Each RAVEN problem consists of a nine-image matrix ($3 \times 3$ grid) where the bottom-right image is removed. The task is to infer the missing image from a set of candidate choices based on the underlying abstract rules. RAVEN contains seven configurations, each containing 10K problems with a specific scene structure. The configurations incorporate five concepts that define problem variations: *Shape* (*e.g.*, triangle, square, pentagon, *etc.*), *Size* (relative scale of objects), *Color* (*e.g.*, white, gray, black, *etc.*) and *Position/Number* (spatial arrangement and number of objects within each image). Each problem follows one or more concept-specific rules that govern changes in these visual concepts: *Constant* (the concept remains unchanged), *Progression*

(the concept changes sequentially), *Arithmetic* (the concept follows a mathematical relationship) and *Distribution-of-Three* (the concept has three permutable values). We use problems from all seven configurations of RAVEN to train and test the models.

**PGM** (Barrett et al., 2018). The PGM dataset is a large-scale RPM-like dataset. The problem structure of PGM problems is similar to RAVEN. A PGM problem also consists of a nine-image matrix, with the bottom-right image removed. The models infer the missing image based on the abstract rules of the given context and choose the correct answer from eight candidate images. The dataset contains 1.4M unique problems. The visual concepts are *shape* and *line* of the *color*, *number*, *position*, *size* and *type*. The visual concepts follow systematically defined rules, which have five categories: *progression*, *XOR*, *OR*, *AND*, and *consistent union*. The PGM dataset provides eight generalization regimes, and we use the neutral regime for model training and evaluation.

### A.2. ID-ZS Tasks

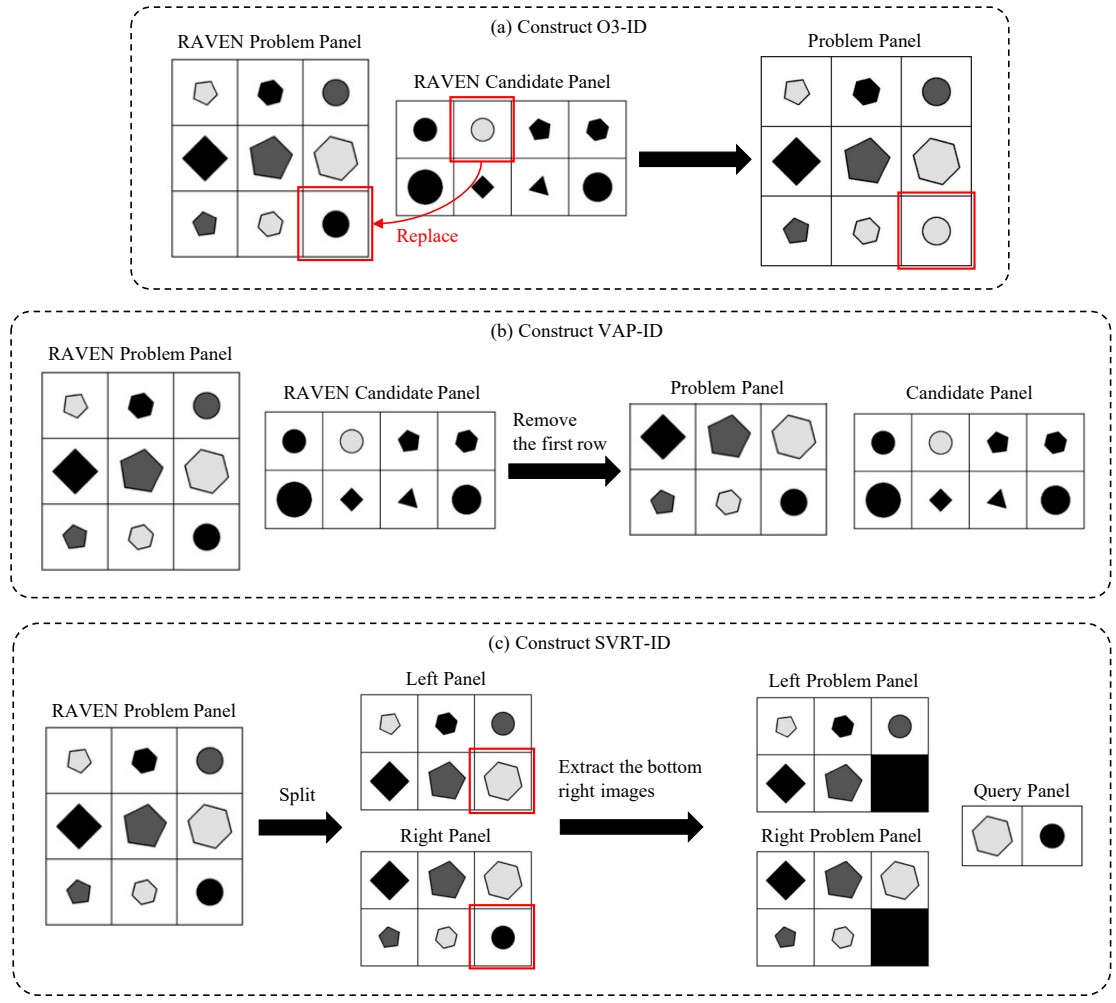

Figure 4: Construction of the ID tasks.

**O3-ID**. As shown in Figure 4a, we replace the bottom-right image of each $3 \times 3$ image panel with a randomly selected distractor from the candidate panel to construct odd-one-out tests from RAVEN. This process transforms the bottom-right image into a rule-breaking image by modifying a specific visual concept of the correct image. The constructed O3-ID dataset shares the same visual concepts and abstract rules as RAVEN but follows the form of odd-one-out, where the models must identify the image breaking the rule of the problem panel.

**VAP-ID**. The construction of the VAP-ID task is relatively straightforward in Figure 4b. Unlike RAVEN, a VAP presents a $2 \times 3$ image matrix as its problem panel, and the abstract rules can change different visual concepts in different rows. Here, we construct VAP problems using RAVEN's design of analogy. Specifically, we remove the final row and retain the first two rows (*i.e.*, the first six images) of each RAVEN problem panel to generate a problem of VAP-ID. In this way, VAP-ID problems preserve the same in-distribution visual concepts and abstract rules as RAVEN, while adopting a different structure of problem panels. These problems are designed to assess whether models can still parse the learned abstract rules and complete the task when the structure of problem panels is changed.

**SVRT-ID**. In Figure 4c, we split each RAVEN problem panel into two parts to construct SVRT-style problems. The first two rows form the left panel, and the last two rows form the right panel. We then extract the bottom-right image from both panels to form the query panel, while the remaining images are kept as the corresponding problem panels. In this way of construction, the left and right problem panels exhibit different abstract rules of the same type. Therefore, the association between the query images and the problem panels is typically unambiguous.

### A.3. OOD-ZS Tasks

**G1-set** (Mańdziuk & Żychowski, 2019). The G1-set dataset evaluates the AVR ability of models through odd-one-out problems. Unlike RPMs, where the task is to infer missing images, an odd-one-out problem requires the models to recognize the oddest figure in a panel based on relational differences in visual concepts. In G1-set, the odd figure can differ on the visual concepts *size*, *shape*, *shading*, *rotation*, *etc*. To ensure the problems are well-defined, the G1-set problems are constructed using a strict rule. In each problem with $N$ figures, there is one subset of $N-1$ figures that share a common set of $1 \sim 3$ features. The remaining figure is the odd image that lacks this commonality. The difficulty of problems is controlled by the number of figures and the number of shared features.

**VAP** (Hill et al., 2019). The VAP dataset evaluates the analogy-making ability of models. Inspired by previous human and machine reasoning tasks like RPMs, VAP requires the models to identify abstract rules and apply them across different visual concepts. Each problem panel consists of a source sequence and a target sequence. The source sequence has three images where an abstract rule is instantiated. The target sequence has two images where the removed position must be completed according to the rule of the source sequence. The candidate panel provides four candidate answer images, including one correct answer and three distractors. The goal is to select the correct answer that best completes the target sequence by analogy with the source sequence. The core challenge in VAP is to apply learned rules across different visual domains. VAP defines four logical or mathematical abstract rules: *XOR*, *OR*, *AND* and *progression*. The abstract rules can be instantiated in seven different visual concepts: *line type*, *line color*, *shape type*, *shape color*, *shape size*, *shape quantity* and *shape position*. Each visual concept contains 10 possible values to increase the complexity of problems.

**SVRT** (Fleuret et al., 2011). The SVRT dataset evaluates the relational visual reasoning ability of models. Unlike traditional image classification tasks that rely on texture, SVRT problems require holistic pattern recognition based on spatial relationships among randomly generated abstract shapes. Each of the 23 problems in the dataset requires assigning a query image to one of two panels, where the distinction is defined by an abstract rule governing the global arrangement of parts. Importantly, these rules are not based on the appearance, location, or topology of individual parts but on how multiple parts interact as a whole. The problems in SVRT involve relational concepts rather than low-level features. Some examples of reasoning principles used in the dataset include: *proximity* (are two or more shapes close to each other), *similarity* (do multiple shapes share the same size/form), *symmetry* (are the parts arranged symmetrically), *topology* (are certain shapes enclosed within others), *counting* (does the image contain an even or odd number of objects) and *identity relations* (do two objects have the same shape or orientation). For example, one problem might require distinguishing two identical shapes vs. two different images.

## B. Details of Models

### B.1. UCGS-T

This section describes the architectures of learnable networks in UCGS-T and the choice of hyperparameters. We introduce the networks in the order of image encoder, image decoder, patch encoder, panel encoder and patch decoder.

- **Image Encoder**:
    - $4 \times 4$ Conv, stride 2, padding 1, 64, ReLU

- $4 \times 4$ Conv, stride 2, padding 1, 64, ReLU
- $4 \times 4$ Conv, stride 2, padding 1, 128, ReLU
- $3 \times 3$ Conv, stride 1, padding 1, 128
- ResBlock, hidden 32, 128
- ResBlock, hidden 32, 128, ReLU
- $4 \times 4$ Conv, stride 2, padding 1, 64, ReLU
- $4 \times 4$ Conv, stride 2, padding 1, 128, ReLU
- $3 \times 3$ Conv, stride 1, padding 1, 128
- ResBlock, hidden 32, 128
- ResBlock, hidden 32, 128, ReLU
- $1 \times 1$ Conv, stride 1, 64

- **Image Decoder**:

    - $3 \times 3$ Conv, stride 1, padding 1, 128, ReLU
    - ResBlock, hidden 32, 128
    - ResBlock, hidden 32, 128, ReLU
    - $4 \times 4$ Deconv, stride 2, padding 1, 64, ReLU
    - $4 \times 4$ Deconv, stride 2, padding 1, 128
    - ResBlock, hidden 32, 128
    - ResBlock, hidden 32, 128, ReLU
    - $4 \times 4$ Deconv, stride 2, padding 1, 64, ReLU
    - $4 \times 4$ Deconv, stride 2, padding 1, 64, ReLU
    - $4 \times 4$ Deconv, stride 2, padding 1, 3

- **Patch Encoder**: The linear network to encode position information is

    - LayerNorm, 128
    - Fully Connected, 128, ReLU
    - Fully Connected, 128

    The patch encoder is a 12-layer Transformer decoder where the hidden size is 128 and the number of attention heads is 8.

- **Panel Encoder**: The panel encoder is a 12-layer Transformer decoder with the hidden size 128 and the number of attention heads 8. We use a linear layer to convert the input keys and values to hidden vectors. Another linear layer is introduced to convert the input queries to hidden vectors.

- **Patch Decoder**: The panel encoder is a 12-layer Transformer decoder with the hidden size 128 and the number of attention heads 8. We use a linear layer to convert the input keys and values to hidden vectors. Another linear layer is introduced to convert the input queries to hidden vectors. The output of the Transformer decoder is projected to the index of the discrete vector in the codebook $e$ by a linear layer without the bias parameter.

The image encoder and image decoder are first trained by setting the learning rate as $4 \times 10^{-4}$ and batch size as 64, referring to the original configuration in VQVAE (Van Den Oord et al., 2017). Then we freeze the parameters of the image encoder and image decoder, training the remaining part by setting the learning rate as $3 \times 10^{-4}$, batch size as 128. We monitor the performance of UCGS-T on the validation set after each training epoch and save the checkpoint with the best validation accuracy. The parameters are updated by Adam optimizer (Kingma & Ba, 2014).

### B.2. Ablation Baselines

Besides the patch-based backbone of UCGS-T, we introduce another two backbones. The object-centric backbone can distinguish objects in scenes and extract representations for individual objects. We adopt the encoder of STSN (Mondal et al., 2023) as the object-centric backbone, and train the backbone using the official code[1]. The monolithic backbone encodes a

---

[1] https://github.com/Shanka123/STSN/tree/main

scene into one representation. The encoder and decoder of the monolithic backbone consist of continuous convolutional layers that map the image into a single representation of size 512. We call the slots and representations extracted from the object-centric and monolithic backbones as patches for convenience. We introduce two types of baseline conditional generative solvers: the Transformer-based solver (Vaswani et al., 2017) and the ANP-based solver (Kim et al., 2019). All the parameters are updated by Adam optimizer (Kingma & Ba, 2014).

**Transformer-based Solver** (Vaswani et al., 2017). The Transformer-based conditional generative network inputs all context patches $\boldsymbol{Z}^c = \{\boldsymbol{Z}_i^c | i = 1, \ldots, N_c\}$ to a Transformer encoder. The output of the Transformer decoder is the key and query of a Transformer decoder, and the embeddings of the target positions $\{\text{PE}_i | i = 1, \ldots, N_t\}$ are regarded as the query of the Transformer decoder. The generative process is

$$
\begin{aligned}
\boldsymbol{h}_i^{kv} &= f_{\text{KV}}\left(\boldsymbol{Z}_i^c\right), & i &= 1, ..., N_c, \\
\hat{\boldsymbol{h}}^{kv} &= \text{TFEncoder}\left(\boldsymbol{h}^{kv}\right), \\
\boldsymbol{h}_i^q &= f_{\text{Q}}\left(\text{PE}_i\right), & i &= 1, \ldots, N_t, \\
\hat{\boldsymbol{h}}^q &= \text{TFDecoder}\left(\boldsymbol{h}^q, \hat{\boldsymbol{h}}^{kv}\right), \\
\boldsymbol{Z}_i^t &= f_{\text{O}}\left(\hat{\boldsymbol{h}}_i^q\right), & i &= 1, \ldots, N_t.
\end{aligned}
\tag{10}
$$

In the generative process, $f_{\text{KV}}$, $f_{\text{Q}}$, and $f_{\text{O}}$ are single-layer feedforward networks. We set the hidden size of the Transformer encoder and decoder as 512, the number of Transformer layers as 12, and the number of attention heads as 8.

**ANP-based Solver** (Kim et al., 2019). The ANP conditional generative baseline regards the mapping from context patches to target patches as stochastic functions. The context slots are used to construct the context of stochastic functions, and the model will sample a function from the posterior to map the embeddings of target positions into target patches. We set the size of the global latent as 512, the number of attention layers as 12, and the number of attention heads as 8, and other hyperparameters and the model architecture follow the 2D regression configuration in (Kim et al., 2019).

### B.3. Task-specific Solvers

We use the official codes of PrAE[2], NVSA[3], GCA[4], ALANS[5] and RAISE[6] in the experiments. All task-specific solvers are trained without additional annotation information. To stabilize the training process, ALANS is trained based on the pretrained checkpoint provided by the authors.

### B.4. GPT-4o

We design task-specific prompts for GPT-4o, which are illustrated in Figure 5. In the prompts, we will provide the information about the problem structures, *e.g.*, the problem panel of PGM is a $3 \times 3$ matrix and the problem panel of VAP is a $2 \times 3$ matrix. Each panel is regarded as an input image of GPT-4o.

### B.5. Computational Resource

All the models are trained on a single 24GB NVIDIA GeForce RTX 4090 GPU and implemented using PyTorch (Paszke et al., 2019).

## C. Additional Experimental Results

### C.1. Answer Generation

We provide additional examples of answer generation. Figures 6 and 7 show the generated results for various AVR problems on the RAVEN and PGM datasets, respectively. Overall, the generated results on RAVEN are more accurate, corresponding

---

[2]https://github.com/WellyZhang/PrAE

[3]https://github.com/IBM/neuro-vector-symbolic-architectures-raven

[4]https://github.com/nivPekar/Generating-Correct-Answers-for-Progressive-Matrices-Intelligence-Tests

[5]https://github.com/WellyZhang/ALANS

[6]https://github.com/FudanVI/generative-abstract-reasoning/tree/main/raise

to the selection accuracy, primarily because the problems of RAVEN contain less noise, which has been discussed in previous works. UCGS-T achieves the best generation performance on both RAVEN and PGM. UCGS-T can handle the uncertainty caused by noise effectively. For example, in the 4th sample of PGM, the shape and position of the generated object differ from the ground truth. TF-Patch generates relatively accurate results on RAVEN, but it tends to produce rule-violating answers on PGM. Overall, TF-Patch generates fewer artifacts in the images. Baselines using the object-centric backbone (TF-OCL and ANP-OCL) struggle to generate clear images, often producing large black regions. Baselines based on the monolithic backbone tend to produce samples that deviate significantly from the ground truths.

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

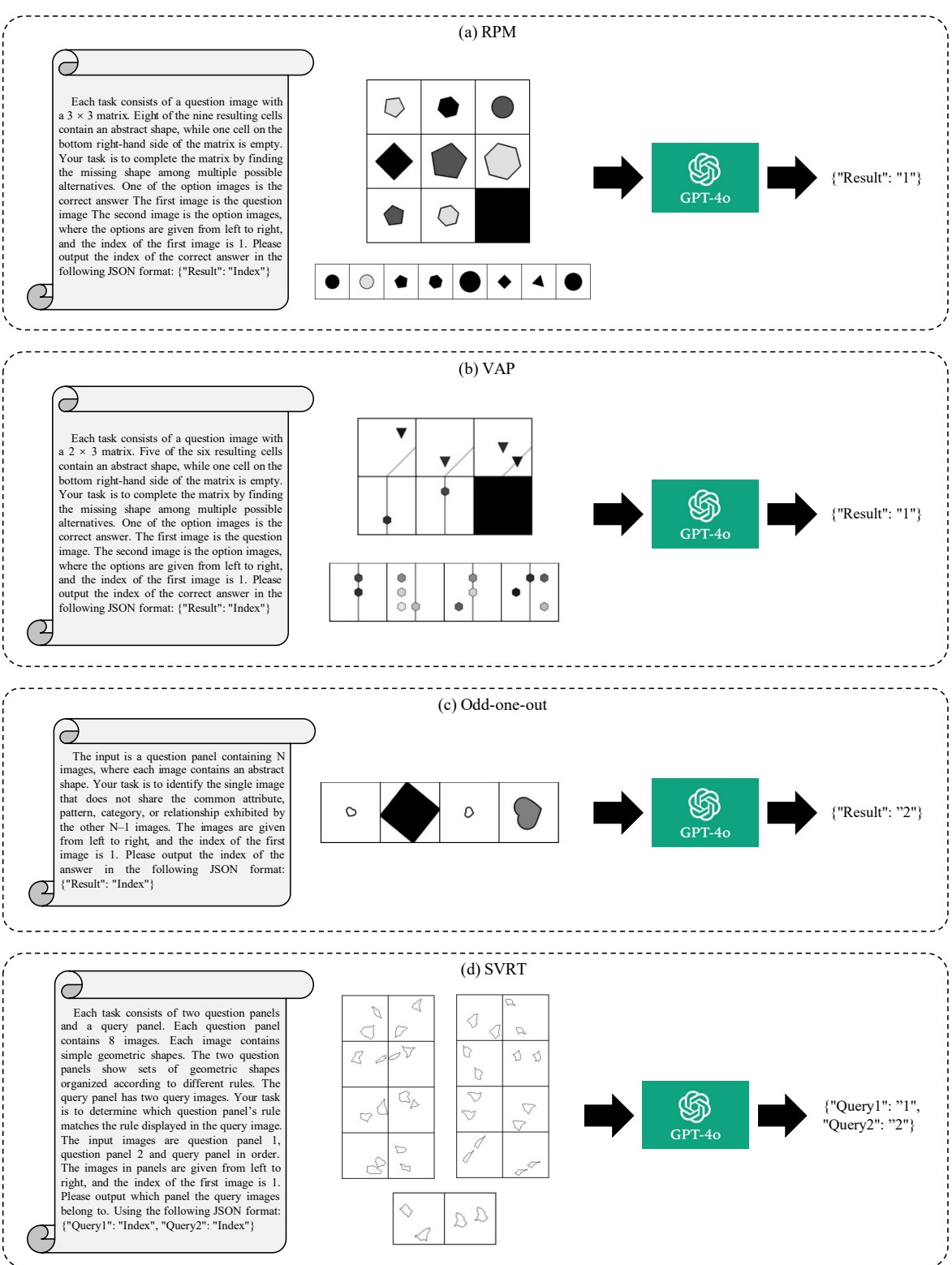

Figure 5: Task-specific prompts of GPT-4o.

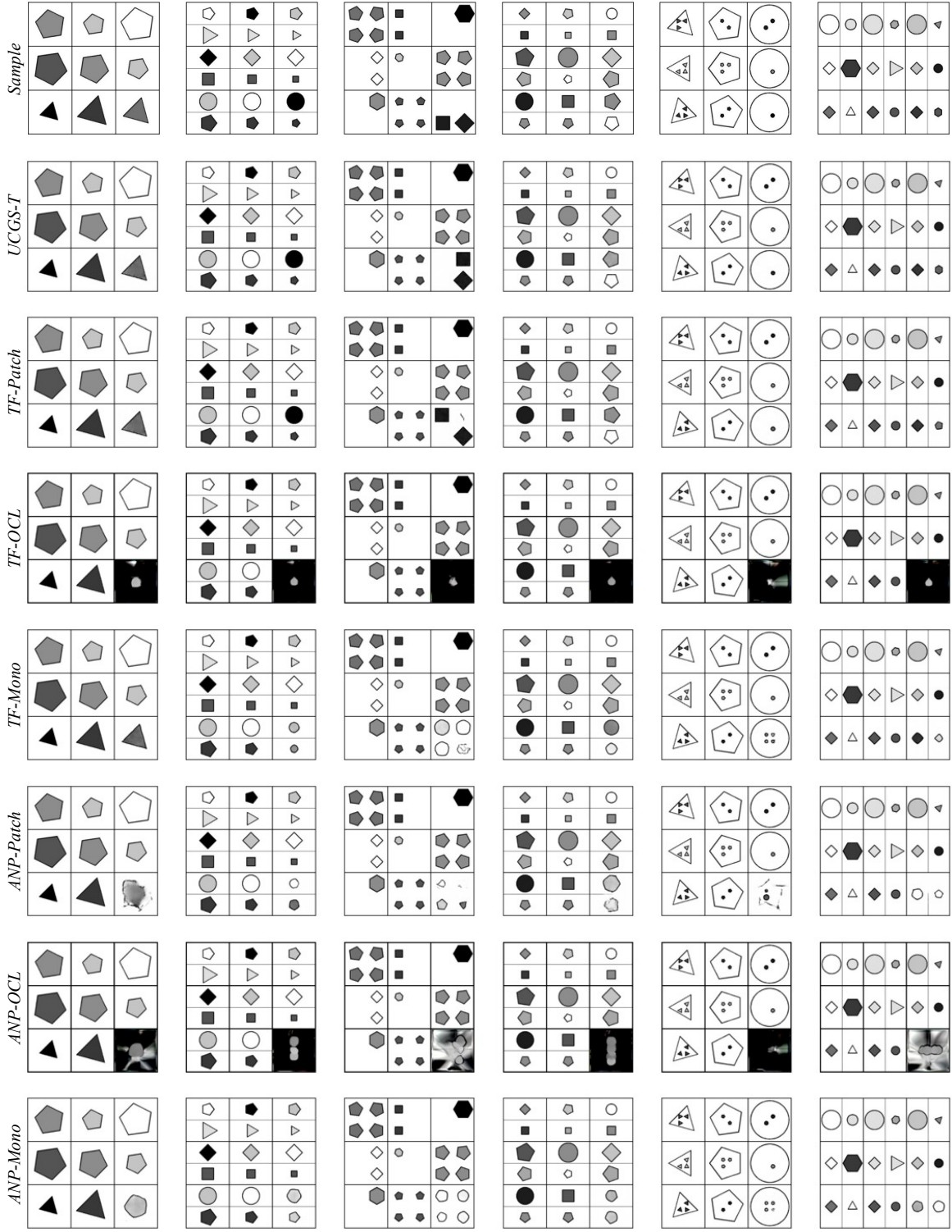

Figure 6: Comparison of generation results on RAVEN.

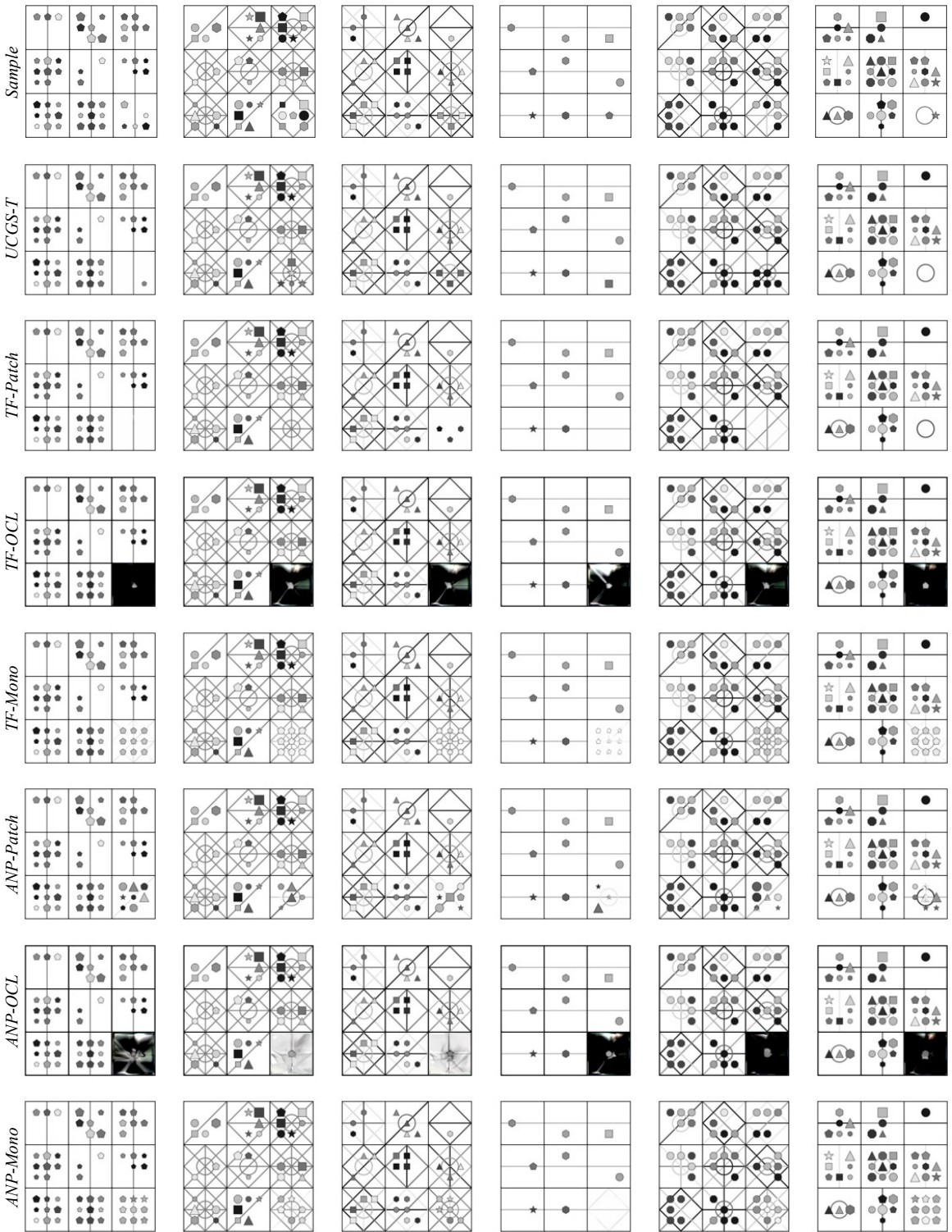

Figure 7: Comparison of generation results on PGM.

