# OpenReview forum: "Beyond Task-Specific Reasoning: A Unified Conditional Generative Framework for Abstract Visual Reasoning"
_ICML.cc/2025/Conference — ICML 2025 poster_

### Official Review · Reviewer_r4C7 · 2025-03-06

**Overall Recommendation:** 3

**Summary:**

- This paper proposes a unified framework (UCGS) for solving 4 different abstract visual reasoning tasks with a single deep network architecture.
- The already existing abstract visual reasoning tasks are based on problem panels consisting of several images showing simple visual concepts following different abstract rules with the model's task being the selection of the correct missing image (Raven's Progressive Matrices, Visual Analogy Problems), the detection of an outlier w.r.t. the rule (Odd-One-Out), or the assignment of images to the correct problem panel (Bongard problems).
- The authors introduce a formal generative framework that cuts these different tasks down to the estimation of conditional probabilites of images given the rest of the problem panel.
- The paper provides a concrete implementation of the framework in form of a transformer architecture applied on vector quantized image patches and consisting of a hierarchy of modules to first encode patches per image, concepts on panel level, and finally generate a target image autoregressively patch by patch.
- An experimental evaluation validates the effectiveness of the proposed framework and architecture in solving various AVR problems without retraining as well as generalization to unseen problems and tasks by showing results for three different settings with:
  - known tasks with unseen combinations of known abstract rules and visual concepts (ID Tasks),
  - unseen tasks with known abstract rules and visual concepts (ID-ZS), and
  - unseen tasks with unseen abstract rules and visual concepts (OOD-ZS).

**Claims And Evidence:**

- The authors claim that "UCGS can successfully solve various AVR tasks" (lines 038 ff.) and the "framework can solve tasks like RPM, Visual Analogy Problem, and Odd-one-out" (lines 094 ff.).
  - While the paper shows that the formal framework is flexible and can be used to tackle these tasks, the experimental results are still far away from completely solving these tasks. Therefore, this statement could be misunderstood and the paper would benefit from a more precise formulation.

- The paper claims "strong reasoning ability in ID tasks" (lines 365 ff.).
  - If I am not mistaken, the paper is missing a comparison with task-specific solvers that would put the documented experimental results in relation to what is possible with inductive biases specific to the tasks and answer the question of the cost of the unified framework and its advantage w.r.t. task flexibility.
  - The provided baselines are rather ablations of the architecture implementing the general framework.

The remaining claims are supported by clear and convincing evidence, e.g., the zero-shot generalization of the reasoning ability to unseen tasks, novel rules and visual concepts by outperforming the random guess.

**Essential References Not Discussed:**

- The main paper is missing descriptions and references for the G1-set and SVRT datasets, while the supplementary material provides these.
- As mentioned above, the paper mentions task-specific solvers but does not discuss the provided experimental results in relation to the performance of these task-specific solvers.

Besides the above, I am not aware of any missing essential references.

**Experimental Designs Or Analyses:**

From the paper and supplementary material, the experimental designs seem valid.

**Methods And Evaluation Criteria:**

Yes, the proposed methods and evaluation criteria make sense.

**Other Comments Or Suggestions:**

- Regarding the lack of clarity w.r.t. the used terms (abstract rules, visual concepts, etc.), the paper would benefit from defining them once early on and then using them consistently.
- Lines 64 to 70 (the first two sentences of that paragraph in the left column) sound very repetitive, which seems like an accident.
- The first paragraph in section 3.1. (before definition 3.1) is a bit repetitive w.r.t. the related work.

**Other Strengths And Weaknesses:**

Strengths:
- The paper addresses a very challenging and interesting problem in learning abstract rules from visual data with a unified framework addressing multiple tasks without retraining.
- It is mostly very clearly explained and easy to understand.
  - The introduction, related work, and figure 1 introduce the topic of abstract visual reasoning very well.
  - The unified framework (section 3.1) and transformer implementation (section 3.2) are explained clearly without being too detailed (hyperparameters etc. in the appendix).
- The generalization capabilities to unseen tasks are exciting and unique to the paper with its unified framework.

Weaknesses:
- Some lack of clarity
  - Is the model training of VQVAE and transformer done end-to-end or as usual one after another? The total loss in eq. 9 seems to suggest the former, which I would find surprising.
  - The paper uses the terms "visual concepts, rules, definitions, problems, and tasks" without a clear definition. The meanings and differences became clear to me rather late in the paper (in sections 4.1.1 to 4.1.3).
  - If I am not mistaken, the provided baselines are rather architecture ablations than existing approaches. If that is the case, the paper would benefit from making this clear.
  - What is the motivation of including an object-centric baseline?

**Questions For Authors:**

The most important questions are:
1. Why do you not compare with task-specific solvers to put the provided experimental results in relation to them, if you mention that "further exploration is required to reach the performance" of them (see limitations lines 431 ff.)?
  - This comparison is very important in my opinion for the evaluation to be complete. Worse performance of the proposed approach is to be expected and would therefore not change my positive view of the experimental results (if not too large).
2. Please see my doubts regarding proposition 3.6. in review section "Theoretical Claims": Does the replacement and therefore removal of an image in the Bongard problem planels not affect the uniqueness of solutions and the probabilities in the proof?
  - A convincing clarification would remove my doubts regarding this theoretical claim.

**Relation To Broader Scientific Literature:**

I am not very familiar with the related works in the area of abstract visual reasoning. However,
- the key contribution being the unified framework to tackle different tasks with a single model without retraining and enabling some generalization to unseen tasks and problems (combinations of abstract rules and visual concepts) seems novel,
- therefore, the generalization ability to unseen tasks and problems is a finding unique to this paper,
- as mentioned in the review section "Claims and Evidence", the paper misses the comparison with the performance of task-specific solvers, which do exist as mentioned in the introduction (lines 37ff. right column) and seem to perform better, as stated in the limitations "further exploration is required to reach the performance achieved by the task-specific solvers" (lines 431ff. right column).

**Theoretical Claims:**

I have checked all theoretical claims and their proofs (propositions 3.4 to 3.6).
- Regarding proposition 3.6., I have some doubts regarding the replacement of the last image in the panel:
  - The Bongard problem only describes the task of assigning a query image to the correct panel (left or right), not replacing an image from the panel with the query image.
  - Can the removal of the last image (or any other image) result in the abstract rule of that panel not being unique anymore and in the worst case even making both query images fitting with both partial problem panels such that there is no unique solution anymore?
  - In lines 183 ff., the authors say that if both problem panels (left and right) are sampled uniformly from the dataset, then the probabilities of the partial panels is 1 over the size of the dataset. If I am not mistaken, that should only be correct if you can make some additional assumptions about the dataset, e.g., that there are no problem panels with equal partial panels, if you remove the last image.

---

> ### Author Rebuttal · Authors · 2025-03-31
>
> Thanks for the constructive suggestions. The detailed responses to the reviewer's comments are as follows.
>
> **Q1: Some statements could be misunderstood and the paper would benefit from a more precise formulation.**
>
> Thank you for pointing out the inaccurate statements. We will carefully check the manuscript to modify or remove the statements in the revised version.
>
> **Q2: Comparison and discussion to task-specific solvers**
>
> We conducted additional experiments on the task-specific solvers and multimodal LLMs GPT-4V and GPT-4o. Since UCGS-T can handle both selective and generative problems, we compare it to generative task-specific solvers PrAE [1], NVSA [2], GCA [3], ALANS [4] and RAISE [5] that support answer selection and generation. The task-specific solvers are trained separately on each dataset without the supervision of rule and attribute annotations to ensure consistency with the experimental setup of UCGS-T.
>
> | Models | RAVEN | PGM |
> | --- | --- | --- |
> | PrAE [1] | 13.6 | - |
> | NVSA [2] | 11.5 | - |
> | GCA [3] | 37.3 | 31.7 |
> | ALANS [4] | 50.1 | - |
> | RAISE [5] | 54.5 | 14.0 |
> | GPT-4V | 13.8 | - |
> | GPT-4o | 19.2 | - |
> | GPT-4o + Language Descriptions | 38.8 | - |
> | UCGS-T (Ours) | 64.6 | 38.1 |
>
> Experimental results show that UCGS-T outperforms generative task-specific solvers under the same setup. PrAE and ALANS cannot solve PGM problems since they only define visual concepts and abstract rules of RAVEN. Without additional annotations, task-specific solvers suffer significant performance drops, e.g., PrAE’s accuracy drops from 65.0% (reported by [1]) to 13.6% in our experiments. We exhibit the results of GPT-4V and GPT-4o on RAVEN from [6], where both models show lower accuracy than UCGS-T, ALANS and RAISE. The multimodal LLMs are not designed for abstract visual reasoning, therefore solving such tasks may require further prompt engineering and chain-of-thought design.
>
> [1] Abstract spatial-temporal reasoning via probabilistic abduction and execution.
>
> [2] A neuro-vector-symbolic architecture for solving Raven’s progressive matrices.
>
> [3] Generating correct answers for progressive matrices intelligence tests.
>
> [4] Learning algebraic representation for systematic generalization in abstract reasoning.
>
> [5] Towards Generative Abstract Reasoning: Completing Raven’s Progressive Matrix via Rule Abstraction and Selection.
>
> [6] What is the visual cognition gap between humans and multimodal llms?
>
> **Q3: Doubts regarding the replacement of the last image in Proposition 3.6**
>
> SVRT provides an image set for each rule containing different images generated by computer programs that follow the rule. Each image is assigned to only one panel of Bongard Problems (BPs). If the remaining images of a set are insufficient to form a complete panel, they will be dropped. Therefore, there are no duplicate images between BP panels even if one image is removed from the left and right problem panels.
>
> **Q4: The main paper is missing descriptions and references for the G1-set and SVRT datasets, while the supplementary material provides these.**
>
> Thank you for the suggestion. We will add the descriptions and references in the first paragraph of Experiments.
>
> **Q5: Is the model training of VQVAE and transformer done end-to-end or as usual one after another?**
>
> VQVAE is pretrained before training the remaining modules. We leave the image reconstruction loss in Eq. 9 to make it possible to finetune VQVAE end-to-end in the training stage. But we find that setting \lambda to 0 (i.e., freezing the parameters of VQVAE) is the best choice. Please refer to Appendix B.1 for the detailed descriptions.
>
> **Q6: Regarding the lack of clarity w.r.t. the used terms**
>
> In this paper, tasks refer to different abstract visual reasoning tasks such as RPM and BP. Problems refer to individual problem instances. Definitions describe the form of different tasks. Visual concepts refer to image attributes like object size and color. Rules are changing patterns on attributes (e.g., progressive changes). Thanks for the helpful suggestion to make the terms clear. We will introduce the terms at the beginning of the method section.
>
> **Q7: The motivation of including an object-centric baseline**
>
> Some task-specific solvers [1, 2] have realized abstract visual reasoning with object-centric representations. Therefore, we adopt it as one of the typical approaches to explore the performance of different visual representations in UCGS.
>
> [1] Learning to reason over visual objects.
>
> [2] Systematic visual reasoning through object-centric relational abstraction.
>
> **Q8: Other comments or suggestions about writing**
>
> Thanks for the constructive comments. In the revised version, we will clarify that the baselines are ablations on model architecture when introducing the baselines. And we will also remove the repeated parts (Lines 64-70 and the first paragraph in Section 3.1) in the manuscript.

---

### Official Review · Reviewer_zHeV · 2025-03-12

**Overall Recommendation:** 1

**Summary:**

This paper presents a method for solving abstract visual reasoning tasks that aims to unify previous methods for different types of tasks (e.g. matrix reasoning vs. odd-one-out) and different modes for solving problems (e.g. classification vs. generation). The method is evaluated on various abstract visual reasoning tasks, with an emphasis on the ability to generalize between tasks.

**Claims And Evidence:**

The primary concern with this work is that, despite claiming to present a highly general model, it is overly focused on a very specific subset of idealized abstract visual reasoning problems. In general, abstract visual reasoning tasks have been created because we think that they tell us something important about the more general reasoning capabilities of our models, not because we are interested in solving these tasks per se. Thus, it is unclear why a 'general purpose solver' is needed for this specific subset of tasks. Despite being more general than some previous models in this literature (e.g., those that are specific to classification-based matrix reasoning tasks), the proposed approach is still very specific to a particular type of problem. It assumes that inputs will be presented in a set of discrete panels, that these panels will consist of relatively simple geometric forms, and that the objects will be entirely explainable via a relatively simple set of abstract rules. Thus, the proposed approach is very far from the 'general purpose abstract reasoner' that is promoted in the abstract and the introduction.

The performance of the model is also very poor in some cases. For instance, the iid performance on the RAVEN and PGM benchmarks is well below the performance of several models which are not included as baselines. The introduction touts the zero-shot reasoning abilities of the model, but the zero-shot generalization to new tasks is generally very poor (though somewhat better than the baselines). There is also no discussion or evaluation of the truly general-purpose systems (LLMs, VLMs, reasoning models like o1) that increasingly display strong abilities to solve these sorts of problems, while also not being limited to a very specific problem format.

**Essential References Not Discussed:**

Within the domain of deep learning models designed to solve abstract visual reasoning tasks, the references considered are reasonable. But many of these models are not directly compared with the proposed approach, despite achieving better performance in some settings. There is also very little consideration of reasoning beyond these types of tasks, and very little discussion of other approaches to solving abstract visual reasoning tasks.

**Experimental Designs Or Analyses:**

The experiments are reasonable given the goals of the paper, but they are very limited to a specific type of abstract visual reasoning task.

**Methods And Evaluation Criteria:**

There are many baselines from the literature on abstract visual reasoning in deep learning systems (e.g. architectures designed for problems like RAVEN and PGM) that are not discussed or compared with the proposed approach. The tasks that are considered are also very similar, despite the emphasis of the paper on generality. There are many other abstract visual reasoning tasks (Bongard problems, ARC, visual question-answering tasks, tasks involving reasoning over real-world images) that are important to consider when evaluating a putatively general-purpose abstract visual reasoning system, and that are increasingly solvable by LLMs or VLMs.

**Other Comments Or Suggestions:**

N/A

**Other Strengths And Weaknesses:**

N/A

**Questions For Authors:**

N/A

**Relation To Broader Scientific Literature:**

The paper primarily considers only other work that evaluates deep learning systems on abstract visual reasoning tasks. There is very little connection to other types of reasoning tasks, or the broader space of models that are increasingly able to solve a wide range of tasks.

**Theoretical Claims:**

The propositions and proofs introduced in section 3.1, though correct, seem somewhat unnecessary for what ultimately turns out to be a relatively straightforward generative architecture for solving these types of abstract visual reasoning problems. The definitions introduced in this section also underscore the very specific domain to which the model can be applied (i.e. it is only applicable to problems involving panels governed by specific rules).

---

> ### Author Rebuttal · Authors · 2025-03-31
>
> Thanks for the constructive suggestions. The detailed responses to the reviewer's comments are as follows.
>
> **Q1: Comparison and discussion to RPM solvers and general-purpose systems**
>
> We conducted additional experiments on the general-purpose systems GPT-4V and GPT-4o and classic task-specific RPM solvers. Since UCGS-T can handle both selective and generative problems, we compare it to generative task-specific solvers PrAE [1], NVSA [2], GCA [3], ALANS [4] and RAISE [5] that support answer selection and generation. The task-specific solvers are trained separately on each dataset without the supervision of rule and attribute annotations to ensure consistency with the experimental setup of UCGS-T.
>
> | Models | RAVEN | PGM |
> | --- | --- | --- |
> | PrAE [1] | 13.6 | - |
> | NVSA [2] | 11.5 | - |
> | GCA [3] | 37.3 | 31.7 |
> | ALANS [4] | 50.1 | - |
> | RAISE [5] | 54.5 | 14.0 |
> | GPT-4V | 13.8 | - |
> | GPT-4o | 19.2 | - |
> | GPT-4o + Language Descriptions | 38.8 | - |
> | UCGS-T (Ours) | 64.6 | 38.1 |
>
> **Comparison with task-specific solvers.** The experimental results indicate that under the same experimental setup, UCGS-T outperforms the generative task-specific RPM solvers. Since PrAE and ALANS only define visual concepts and abstract rules for RAVEN, they are incapable of solving PGM problems. Our experiments reveal that, without additional annotation information, the performance of the generative task-specific RPM solvers significantly declines. For example, [1] reports that PrAE achieves accuarcy of 65.0% after rule-supervised training, which declines to 13.6% in our experiments.
>
> **Comparison with general-purpose systems.** We also exhibit the results of the general-purpose systems GPT-4V and GPT-4o on RAVEN. The accuracy scores are reported by [6] where GPT-4V and GPT-4o have lower accuracy than UCGS-T, ALANS and RAISE. GPT-4o shows a notable performance improvement when provided with language descriptions of candidate images (from 19.2% to 38.8%), but still falls short of UCGS-T, ALANS, and RAISE. While the general-purpose systems GPT-4V and GPT-4o are not specifically designed for abstract visual reasoning, solving such tasks may require further prompt engineering and chain-of-thought design.
>
> [1] Abstract spatial-temporal reasoning via probabilistic abduction and execution.
>
> [2] A neuro-vector-symbolic architecture for solving Raven’s progressive matrices.
>
> [3] Generating correct answers for progressive matrices intelligence tests.
>
> [4] Learning algebraic representation for systematic generalization in abstract reasoning.
>
> [5] Towards Generative Abstract Reasoning: Completing Raven’s Progressive Matrix via Rule Abstraction and Selection.
>
> [6] What is the visual cognition gap between humans and multimodal llms?
>
> **Q2: Despite claiming to present a highly general model, it is overly focused on a very specific subset of idealized abstract visual reasoning problems.**
>
> As stated in Lines 26–29 of the abstract, UCGS is "general" since it aims to solve multiple abstract visual reasoning (AVR) tasks in a unified manner, which are often treated as independent in task-specific solvers. We agree that AVR tasks are widely used not because researchers are interested in achieving high scores in these tasks, but because they can reveal the core reasoning abilities of AI. Our goal is precisely to develop a more general framework for AVR—one that can exhibit multi-task AVR ability as humans, rather than simply “solving these tasks.”
>
> To this end, we use representative AVR tasks as benchmarks. While the matrix reasoning tasks may seem simple in form, building multi-task solvers on them remains challenging. As shown in the table, GPT-4V and GPT-4o can hardly achieve performance comparable to task-specific solvers. The performance of UCGS-T illustrate the effectiveness of the proposed framework in the typical AVR tasks. Importantly, selecting these tasks does not mean our framework is limited to them. UCGS can extend to other tasks, such as visual analogy extrapolation problems [1], which shares a similar structure with matrix reasoning.
>
> [1] A review of emerging research directions in abstract visual reasoning.
>
> **Q3: The propositions and proofs in section 3.1 seem somewhat unnecessary …**
>
> The significance of the propositions consists of two aspects. On one hand, it demonstrates how different forms of classical reasoning tasks can be described as a conditional generation processes. On the other hand, it explains how generative and selective tasks can be unified into one framework. Specifically, some selective tasks can be reformulated as an implicit answer generation problem, where the goal is to search for an option among predefined candidates that matches the generated result. This design allows a single conditional generative model to solve both types of tasks simultaneously, eliminating the requirement of an additional scoring network for answer selection.

---

> > ### Comment · Reviewer_zHeV · 2025-04-03
> >
> > Thank you to the authors for these replies. While I appreciate the engagement and additional results, I feel that my core concerns have not been addressed.
> >
> > First, the performance of the model is not particularly competitive relative to previous approaches on these datasets. New baselines have been included to compare with other generative task-specific solvers, but there are many non-generative (classification-based) models that perform much better on these tasks, many of them now approaching saturation on both RAVEN and PGM. I see no principled reason not to include these other baselines in the comparison.
> >
> > Second, despite the purported general-purpose nature of the approach, it is still highly specific to the format of abstract visual reasoning tasks, and it is not even clear how it will be generalized to closely related tasks like ARC, or visual reasoning tasks that involve real-world images. Though I appreciate that the task is slightly more general relative to some other approaches in this literature, it is still very specifically tailored to these kinds of tasks, and there is no explanation of how the approach will be scaled to handle more open-ended, unstructured, real-world reasoning problems, or how it will be integrated into more general-purpose systems. Again, it seems that the purpose is merely to solve abstract visual reasoning tasks for their own sake, rather than to develop an approach that can improve the reasoning abilities of real-world agents.
> >
> > Third, there are new baselines reported for gpt-4v and gpt-4o, but these are not state-of-the-art systems for reasoning. It would be more appropriate to perform a comparison with reasoning models like o1 or r1. Additionally, it is unclear if current systems struggle with visual reasoning tasks because of the reasoning or visual components of the tasks. There is a lot of work suggesting that multimodal models struggle primarily with visual encoding. Therefore, it would be especially informative to disentangle the visual vs. reasoning demands of these tasks when comparing with baselines like gpt-4o or reasoning models like o1.

---

### Official Review · Reviewer_v5Nm · 2025-03-13

**Overall Recommendation:** 3

**Summary:**

The authors transform a series of different classical abstract visual reasoning tasks by making them all into a task of generating one missing data panel given the remaining set of example panels instantiating a visual concept, which is captured with conditional generative models. Using a new architecture based on transformers, the system is trained on several abstract visual reasoning tasks, performs well across tasks, and transfers even to unseen tasks.

**Claims And Evidence:**

While I could not run the code to replicate the experiments, the reported experiments and evaluations support the claims.

**Essential References Not Discussed:**

There are multimodal models that have addressed similar problem settings, e.g.
- Zhao, H., Cai, Z., Si, S., Ma, X., An, K., Chen, L., Liu, Z., Wang, S., Han, W. and Chang, B., MMICL: Empowering Vision-language Model with Multi-Modal In-Context Learning. In The Twelfth International Conference on Learning Representations.
and neuro-symbolic systems, e.g.:
- Hersche, M., Zeqiri, M., Benini, L., Sebastian, A. and Rahimi, A., 2023. A neuro-vector-symbolic architecture for solving Raven’s progressive matrices. Nature Machine Intelligence, 5(4), pp.363-375.
If the Bongard problems are part of this paper, then it seems necessary to cite
- Yun, X., Bohn, T. and Ling, C., 2020. A deeper look at Bongard problems. In Advances in Artificial Intelligence: 33rd Canadian Conference on Artificial Intelligence, Canadian AI 2020, Ottawa, ON, Canada, May 13–15, 2020, Proceedings 33 (pp. 528-539). Springer International Publishing.
- Depeweg, S., Rothkopf, C.A. and Jäkel, F., 2024. Solving bongard problems with a visual language and pragmatic constraints. Cognitive Science, 48(5), p.e13432.

**Experimental Designs Or Analyses:**

The abstract visual reasoning tasks are standard benchmarks.

**Methods And Evaluation Criteria:**

The authors use selection accuracy as a single evaluation criterion.

**Other Comments Or Suggestions:**

Fig. 1 (d) does not show BPs but SVRT by Fleuret et al. (2011).

In fig. 2 (a) it looks like there is a redundant positional embedding 2 /3.

BP is used in proposition 3.6, although the Bongard Problems are introduced later in the text and then disappear altogether.

**Other Strengths And Weaknesses:**

The goal of the study is very relevant, timely, and interesting.

It would be very helpful to contextualize the performance results: how well do multimodal models fair by comparison, and why? How do neurosymbolic models perform, and why?

While I am convinced of the helpfulness of the old approach, “if you cannot solve a problem, solve a different problem”, it should be pointed out that e.g. the Bongard problems are much harder in their original formulation in which a sentence describing the concept has to be formulated in natural language.

**Questions For Authors:**

The proposed architecture is rather intricate and complex. Are there any insights the authors would like to share beyond the performance on the benchmarks?

Can the authors share any reasoning for the specific form of the

Can the authors explain the large improvement in accuracy in the ID tasks with comparatively little improvement compared to baseline in the out of distribution tasks?

**Relation To Broader Scientific Literature:**

Abstract visual reasoning is of broad interest to the community. The present paper profits most from reformulating classic problems into one single problem, i.e., generating a new data point (panel) based on a number of positive examples (panels) from the same concept. While this is an interesting manipulation that allows the transfer across different AVR tasks, it is also a strong modification of the original problems, e.g. in the case of the Bongard problems, where the original goal is to produce a sentence describing the concept of one set versus the concept of a second set.

Conceptually, it is an important goal to learn more abstract visual concepts, but it is not clear how the current system that learns such implicit abstract representations would compare to neuro-symbolic systems, particularly in terms of explainability.

Intuitively, it is unclear why the performance on the RPM task is comparatively low to other systems trained exclusively on this task and why out of distribution generalization is comparatively to the other systems.

**Theoretical Claims:**

The paper does not contain any proof. Propositions 3.4 to 3.6 are rather basic intuitions that are translated from language to straightforward formulas.

---

> ### Author Rebuttal · Authors · 2025-03-31
>
> Thanks for the constructive suggestions. The detailed responses to the reviewer's comments are as follows.
>
> **Q1: Concerns about the formulation and experiments of Bongard problems**
>
> Similar to the task-specific solvers, UCGS is a framework designed to solve abstract visual reasoning (AVR) tasks with only visual input/output. Therefore, we adopt a simplified version of Bongard Problems (BPs) mentioned in [1], where the task of describing rules via natural language is transformed into a classification problem on query images. Since the modified BPs have a similar definition to SVRT, we validate the ability of models to solve the modified BPs based on SVRT in our experiments. We will provide a more detailed explanation in the revised manuscript to clarify the setting of BPs.
>
> [1] A review of emerging research directions in abstract visual reasoning.
>
> **Q2: Comparison and discussion to multimodal models and neuro-symbolic models**
>
> We conducted additional experiments on multimodal LLMs GPT-4V and GPT-4o and task-specific solvers. Since UCGS-T can handle both selective and generative problems, we compare it to generative task-specific solvers PrAE [1], NVSA [2], GCA [3], ALANS [4] and RAISE [5] that support answer selection and generation, where PrAE and NVSA are neuro-symbolic models. The task-specific solvers are trained separately on each dataset without supervision of rule and attribute annotations to ensure consistency with the experimental setup of UCGS-T.
>
> | Models | RAVEN | PGM |
> | --- | --- | --- |
> | PrAE [1] | 13.6 | - |
> | NVSA [2] | 11.5 | - |
> | GCA [3] | 37.3 | 31.7 |
> | ALANS [4] | 50.1 | - |
> | RAISE [5] | 54.5 | 14.0 |
> | GPT-4V | 13.8 | - |
> | GPT-4o | 19.2 | - |
> | GPT-4o + Language Descriptions | 38.8 | - |
> | UCGS-T (Ours) | 64.6 | 38.1 |
>
> The experimental results indicate that under the same experimental setup, UCGS-T outperforms the generative task-specific solvers.
>
> **Comparison with neuro-symbolic models.** Without additional annotations, neuro-symbolic models show a significant performance drop on RPMs. [1] reports PrAE achieves 65.0% accuracy with rule supervision, but it drops to 13.6% in our experiments. PrAE and NVSA rely on predefined rules and explicitly defined representations for specific datasets, making it hard for reasoning with undefined concepts and rules. UCGS-T does not rely on manually designed concepts and rules. It performs independent reasoning process on each visual concept, inferring concept and rule without additional annotations.
>
> **Comparison with GPT-4V and GPT-4o.** We exhibit GPT-4V and GPT-4o's performance on RAVEN from [6]. Both models have lower accuracy than UCGS-T, ALANS, RAISE and GCA. GPT-4o improves with candidate image descriptions but remains behind UCGS-T, ALANS, and RAISE. Since not designed for abstract visual reasoning, GPT-4o may require further prompt engineering and chain-of-thought design to improve the performance on RPM.
>
> [1] Abstract spatial-temporal reasoning via probabilistic abduction and execution.
>
> [2] A neuro-vector-symbolic architecture for solving Raven’s progressive matrices.
>
> [3] Generating correct answers for progressive matrices intelligence tests.
>
> [4] Learning algebraic representation for systematic generalization in abstract reasoning.
>
> [5] Towards Generative Abstract Reasoning: Completing Raven’s Progressive Matrix via Rule Abstraction and Selection.
>
> [6] What is the visual cognition gap between humans and multimodal llms?
>
> **Q3: Missing references to some papers of multimodal models and Bongard problems**
>
> Thanks for the suggestions. We will cite them in the revised version.
>
> **Q4: Can the authors explain the large improvement in accuracy in the ID tasks with comparatively little improvement compared to baseline in the out of distribution tasks?**
>
> The performance of models in ID tasks is influenced by the architecture and composed modules. In OOD tasks, the difference between the training data and unseen visual concepts or abstract rules dominate the performance of models, which may reduce the performance difference between baselines and UCGS-T.
>
> **Q5: The proposed architecture is rather intricate and complex. Are there any insights the authors would like to share beyond the performance on the benchmarks?**
>
> UCGS-T extracts global visual concepts from patch-based representations for reasoning. RPM tests often involve global rules, e.g., consistent object counts across panels, which patch-based and object-centric representations struggle to capture. UCGS-T addresses this by adding a module to extract and reason independently on visual concepts. Exploring better image tokenizers may help build models with stronger OOD reasoning capabilities.
>
> **Q6: In fig. 2 (a) it looks like there is a redundant positional embedding 2 /3.**
>
> The positional embedding 2 indicates the target position and is used as the query vector in the panel encoder.

---

### Decision · Program_Chairs · 2025-05-01

**Decision:**

Accept (poster)

**Comment:**

The authors present a novel framework for solving visual reasoning tasks based on conditional generative models. The work was received with mixed scores by the reviewers. As positive aspects, they highlighted the goal and novel approach of the paper, aiming to generalize across tasks. Negative aspects include skepticism regarding the general direction of the paper, sitting between specialized classification approaches and very general reasoning methods based on LLMs/VLMs. Also, additional comparisons against neuro-symbolic methods and LLMs were requested.

In the rebuttal, the authors followed up with the requested comparisons, satisfying two out of three reviewers (Weak Accept). One reviewer remained unconvinced (Reject), raising general concerns about the direction of the paper in discussions, specifically, that it is neither fully general (in compared to LLM/VLM reasoning) nor does it beat fully specialized classification methods.

In light of the diverging situation, I also took a look at the paper. I believe that statements made by all three reviewers are correct and that the decision comes down to how much value we put on novel exploration if the results do not outperform all existing paradigms. As I strongly believe that such exploration is of value to the community, that this is a well written and executed paper, and I believe the general direction to be promising, I decided to follow the two positive reviewers and recommend to accept this work.